# TABDPT: SCALING TABULAR FOUNDATION MODELS

## ABSTRACT

The challenges faced by neural networks on tabular data are well-documented and have hampered the progress of tabular foundation models. Techniques leveraging in-context learning (ICL) have shown promise here, allowing for dynamic adaptation to unseen data. ICL can provide predictions for entirely new datasets without further training or hyperparameter tuning, therefore providing very fast inference when encountering a novel task. However, scaling ICL for tabular data remains an issue: approaches based on large language models cannot efficiently process numeric tables, and tabular-specific techniques have not been able to effectively harness the power of real data to improve performance and generalization. We are able to overcome these challenges by training tabular-specific ICL-based architectures on real data with self-supervised learning and retrieval, combining the best of both worlds. Our resulting model – the Tabular Discriminative Pre-trained Transformer (TabDPT) – achieves state-of-the-art performance on the CC18 (classification) and CTR23 (regression) benchmarks with no task-specific fine-tuning, demonstrating the adapatability and speed of ICL once the model is pre-trained. TabDPT also demonstrates strong scaling as both model size and amount of available data increase, pointing towards future improvements simply through the curation of larger tabular pre-training datasets and training larger models.

## 1 INTRODUCTION

Tabular foundation models (TFMs) have recently emerged as a critical area of research (van Breugel & van der Schaar, 2024) given the importance of tabular data in real-world applications. However, the high heterogeneity of tables, low availability of high quality data, and the lack of obvious inductive bias have made it especially challenging to adapt neural architectures to tabular data (Grinsztajn et al., 2022; McElfresh et al., 2023). Consequently, deep learning techniques and TFMs have not been established as the standard for solving discriminative tabular tasks, with tree-based frameworks such as XGBoost (Chen & Guestrin, 2016) or CatBoost (Prokhorenkova et al., 2018) remaining the default. These ap-

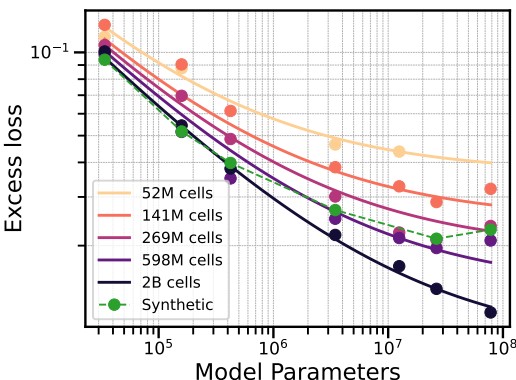

Figure 1: Scaling behaviour for our models. For real data (solid), increasing the model or data size leads to improvements predictable by power laws. Synthetic data (dotted) becomes less useful than real data as the models grow larger. Details are in Section 5.2.

proaches have demonstrated the practical ability to more gracefully handle the idiosyncrasies of tabular data, although they require costly rounds of training and hyperparameter tuning on each new dataset to achieve good results. Indeed, it is unlikely that tree-based models will ever provide training-free generalization to unseen data – which we have grown to expect of foundation models in other domains – and as such we continue to pursue neural approaches, despite the current challenges.

In-context learning (ICL) – referring to the phenomenon where a model generalizes to new tasks using only in-context template examples with no additional fine-tuning – is one avenue showing promise in building neural networks that can dynamically adapt to input data. ICL was first observed in large language models (LLMs) (Brown et al., 2020), which have even demonstrated some ability to perform inference on smaller tabular datasets (Han et al., 2024; Gardner et al., 2024). Since

tables are not text, though, it is challenging to apply LLMs to tabular data. The cell-based, textual tokenization in particular is highly inefficient and makes context size a major limitation (Fang et al., 2024). This has hindered the adoption of LLM-based ICL techniques in practical tabular settings. An alternative technique directly trained to perform ICL is the transformer-based TabPFN (Hollmann et al., 2023), designed specifically for tabular data. TabPFN is pre-trained exclusively on synthetic data (Müller et al., 2022) and is able to more efficiently use its context by avoiding cell-based tokenization: instead, the *rows* essentially act as tokens. While the performance of TabPFN is impressive (McElfresh et al., 2023), especially since the lack of task-specific fine-tuning greatly speeds up inference time, the lack of training on real data leaves something to be desired: we conjecture that the synthetic data generation procedure used in training is not sufficiently diverse, and improving it is a highly non-trivial task. Furthermore, it cannot natively perform regression, and struggles as dataset size increases, greatly limiting its potential as a TFM.

Given the adaptability and efficiency of ICL, we would like to scale it to both handle larger datasets and benefit from real pre-training data. The former can be accomplished using retrieval-based training for more efficient use of the context; Thomas et al. (2024) showed that this can improve performance when fine-tuning on specific tasks, and we demonstrate here that it can also be adapted to the pre-training phase. As for the latter, we can turn to self-supervised learning (SSL) techniques to augment the relatively low amount of quality pre-training tabular data that is publicly available. Specifically, we perform random column prediction to enhance the amount of training data, analogously to what has been done in language (Devlin et al., 2019) and vision (He et al., 2022). Combining a transformer-based ICL with retrieval-based SSL results in our method – the **Tabular Discriminative Pre-trained Transformer** (**TabDPT**) – which demonstrates impressive performance even on brand new tabular tasks. We summarize our contributions below:

1. We introduce TabDPT as a TFM that performs both classification and regression on unseen datasets with *no additional training or hyperparameter tuning*, backed by transformer-based ICL, retrieval-based self-supervised pre-training, and retrieval-based inference.

2. We comprehensively evaluate TabDPT on the OpenML-CC18 (Bischl et al., 2021) and OpenML-CTR23 (Fischer et al., 2023) benchmarks, showing state-of-the-art performance on unseen datasets *even when compared with methods that train on that data with 30 rounds of per-dataset hyperparameter tuning*. Our runtime is therefore also much lower than these baselines once we have a pre-trained model.

3. As there is no single accepted benchmark in the tabular domain, we introduce the idea of using duel-based ranking methods (Elo, 1967; Glickman, 2012) to evaluate the relative performance of models even when pairwise comparison across all datasets is unavailable, mimicking similar developments in LLMs (Chiang et al., 2024).

4. We show that the performance of TabDPT scales with both model size and amount of training data, with Figure 1 in particular demonstrating the power of pre-training with real data.

5. We will release all code, which includes the weights of the trained TabDPT,[1] methods for training TabDPT, a comprehensive evaluation suite, and a library for detecting leakage in tabular datasets which confirms that we are not training on downstream data.[2]

## 2 RELATED WORK

**Tabular Foundation Models (TFMs)** Although TFMs lag behind foundation models in other domains (van Breugel & van der Schaar, 2024), a variety of attempts with different base architectures have emerged. Most similar to ours is TabPFN (Hollmann et al., 2023); TabDPT's architecture relies heavily on this model – albeit with a separate regression head and a retrieval component – but uses a completely different self-supervised pre-training procedure with real data. Both TabDPT and TabPFN are trained to do ICL directly. Another class of ICL-based approaches to build TFMs is to adapt existing LLMs, which can be done, e.g., for discriminative (Hegselmann et al., 2023; Gardner et al., 2024) and generative (Borisov et al., 2022; Wen et al., 2024) tabular tasks. While these techniques can naturally handle textual information in the form of table metadata, column names, and categorical features, they cannot as easily handle the numerical content of tables as we discuss

---

[1] https://github.com/layer6ai-labs/TabDPT

[2] The training, evaluation, and dataset libraries will be released at a later date.

in Section 3.1; this is also borne out in their weaker performance overall (Fang et al., 2024). Finally, there are other tabular-specific architectures, including graph (Kim et al., 2024) and diffusion-based (van Breugel et al., 2024; Lin et al., 2024) techniques for prediction and generation, respectively, showing the ability to also incorporate textual information into the overall modelling pipeline. However, these models are not able to generalize to unseen tasks without supervised fine-tuning.

**Self-Supervised Learning and Generalization in the Tabular Domain**  SSL has proven to be successful for text and images (Devlin et al., 2019; Dosovitskiy et al., 2021), but has not demonstrated the same level of success on tabular data. Many tabular SSL methods cannot generalize beyond the dataset they were pre-trained on (Huang et al., 2020; Yoon et al., 2020; Majmundar et al., 2022; Sui et al., 2024). This raises the question of whether tabular SSL methods can benefit from cross-task training. The answer to this appears increasingly likely to be in the affirmative, as it has been shown very recently that even tree-based methods benefit from tuning their default hyperparameters across tasks (Holzmüller et al., 2024); this same work, following (Rubachev et al., 2022), demonstrates that basic MLPs can also be competitive in predictive tabular tasks when leveraging SSL. Consequently, tabular SSL methods have begun to show generalization across tasks and competitive performance (Zhu et al., 2023; Ye et al., 2023). However, they still require task specific fine-tuning and hyper-parameter tuning, which can be time- and resource-intensive. The only other tabular SSL method we have seen that is able to generalize across tasks without task-specific fine-tuning is by Gardner et al. (2024). However, this 8 billion parameter model still only has a maximum context size of 32 data points – as it is LLM-based – and its performance is not competitive. To our knowledge, we are the first to demonstrate competitive performance and successful generalization of tabular SSL across tasks without task-specific fine-tuning or hyperparameter tuning.

## 3 METHOD

In this section, we outline the architecture of our model, TabDPT, along with the self-supervised learning and retrieval strategies we employ that are key for model performance.

### 3.1 TRANSFORMER ENCODER FOR IN-CONTEXT LEARNING ON TABULAR DATA

Our main goal in this work is to understand how to build tabular foundation models that will scale with model size and amount of data. First, we focus on the architecture. We have found the backbone of TabPFN (Hollmann et al., 2023) to be suitable: it is a non-autoregressive transformer encoder wherein entire *rows* of incoming tabular data can attend to each other and thus play the role of "tokens". More precisely, for every input table with $N$ rows and $F$ features, we first standardize the feature dimension to a fixed size $F_{\max}$, achieved by either padding with zeros or subsampling features. The table is then embedded into a tensor of shape $(N, d)$, where $d$ represents the transformer's hidden dimension, via a linear layer. We do not handle categorical or numerical variables differently, with more details on that in Section 3.2. Subsequently, in the transformer layers, we treat the row dimension $N$ as the sequence length so that individual instances can attend to each other.

A row-based tabular encoding contrasts with that of LLMs which require tokenization of each cell in the input data, inflating the memory requirements by a factor of $F \times \langle N_{\text{tok}} \rangle$, where $\langle N_{\text{tok}} \rangle$ is the average number of tokens per cell. Even with techniques such as sparse attention (Child et al., 2019) and Byte Pair Encoding (Gage, 1994), the overhead remains significant, limiting the table size that can be processed. A similar phenomenon is observed in the image domain when comparing pixel-based transformers (Chen et al., 2020) to ViTs (Dosovitskiy et al., 2021) which use coherent numerical patches of images for embedding, resulting in more efficient models. Similarly, we argue that tables should be divided into *structurally meaningful units*, such as rows, for more efficient processing.

By reducing memory consumption, row-based encoding permits processing a large number of rows, which in turn enables efficient ICL on new tables. In contrast, cell-based methods (Huang et al., 2020; Gorishniy et al., 2024; van Breugel et al., 2024) are limited in the number of rows they can process at a time, rendering them incapable of processing full tables. In turn this prevents ICL and requires additional training or fine-tuning on new datasets, unlike our method.

The final point to discuss on architecture is our approach for training both classification and regression with the same backbone, which is also the biggest change from the architecture of Hollmann et al. (2023). For this, we attach two heads after the transformer layers, each consisting of two-layer

MLPs. For classification, the outputs are logits of a predetermined maximum number of classes,[3] trained using the cross-entropy loss. For regression, the output is a scalar and we simply train using the mean-squared error (MSE) loss. We investigated recasting regression as classification, as in previous work (Imani et al., 2024; Farebrother et al., 2024), but MSE was more effective for us. The full architecture is depicted in Figure 2b, with additional details provided in Appendix H.

## 3.2 Self-Supervised Learning on Tabular Data

Although most of the datasets we use for training are labelled datasets containing pairs of input data $X$ and the corresponding targets $y$, we use a purely self-supervised approach that does not treat $y$ differently from any other feature of $X$; we treat all datasets as unlabeled. We do so to increase the number of relationships between the features that we can learn. We take inspiration from the masked modelling objectives popularized in vision (Germain et al., 2015; He et al., 2022) and language (Devlin et al., 2019); namely, we aim to predict one feature from a random subset of the other features. In more detail, this involves two complementary steps.

---

**Algorithm 1** Training Step of TabDPT

1: Select $B$ random datasets $\{\mathcal{D}_i\}_{i=1}^{B}$
2: **for** each dataset $\mathcal{D}_i$ **do**
3:     Generate $y_i$ from a random column $c_i$.
4:     Sample $N$ close points with $c_i$ dropped.
5:     Shuffle and pad them to obtain $X_i$.
6: **end for**
7: Stack $\{X_i\}_{i=1}^{B}$ and $\{y_i\}_{i=1}^{B}$ into $X$ and $y$.
8: Randomly divide $y$ into $y_{\text{ctx}}, y_{\text{qy}}$.
9: input, target $\leftarrow [X \equiv [X_{\text{ctx}}, X_{\text{qy}}], y_{\text{ctx}}], y_{\text{qy}}$
10: Calculate loss and perform model update.

---

**Random Column as Target** We randomly select a column from the tabular dataset that satisfies specific criteria, such as having a sufficient number of unique values, and treat it as a target for either classification or regression tasks. This allows the model to learn useful representations from the data without relying on external labels. For regression, we simply standardize the values of that column. For classification, if the number of unique values is high, we distribute the values over random partitions and use those as target classes.

**Column Shuffling and Masking** We also shuffle the order of columns, and drop some, to encourage learning robust relationships between features independent of their positional arrangement, improving the model's ability to generalize to new datasets with different feature arrangements.

By combining these self-supervised learning techniques with our transformer architecture, we create a model that is not only robust and scalable, but also capable of learning from all features of any dataset. This allows us to learn efficiently from a limited number of training datasets, whereas only learning from the supervised target would not contain enough learning signal for the model (cf. Figure 4b). Detailed pseudo-code can be found in Code Block 1 and Code Block 2 in the Appendix.

## 3.3 Training with End-to-end Retrieval

One of the primary challenges in training in-context transformer-based models for tabular data is the quadratic growth of compute and memory usage with the context length. This limitation restricts the number of support examples that can be effectively utilized within the context window. While language-based models (Gardner et al., 2024) or TabPFN (Hollmann et al., 2023; McElfresh et al., 2023) can handle small datasets where the entire training set can fit within the context, their scalability to larger, more complex datasets is limited. This raises the question of how to efficiently select and use context when dealing with larger datasets.

Recently Thomas et al. (2024) and Xu et al. (2024) showed that using a dynamic context local to each query point greatly improves the performance and scalability of TabPFN at inference. We hypothesized that training our model end-to-end on local context would improve the downstream performance even further as it results in a better alignment between training and testing objectives.

This objective is similar to works such as RETRO (Borgeaud et al., 2022) or TabR (Gorishniy et al., 2024) which have shown improved results by using retrieval during training. However, performing an exact $k$NN search for each point in the minibatch during training is expensive. The main cost is not from the search itself, but rather the fact that now each query point has its own unique context,

---

[3] Although we show in Section 3.4 how to lift this restriction.

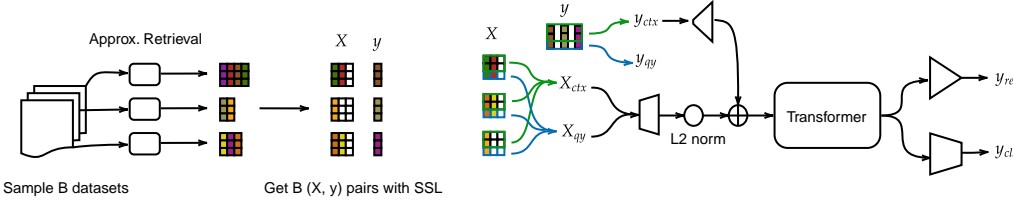

(a) Selecting a training batch  (b) Overview of the architecture

Figure 2: Global overview of our method. (a) We sample different tables from different datasets within a batch. From this, we construct a tensor $X$ of shape $(B, N, F_{\max})$ and target $y$ of shape $(B, N)$ containing class labels or regression targets. (b) We split those tensors along the second (row) dimension to build our input $[X \equiv [X_{\text{ctx}}, X_{\text{qy}}], y_{\text{ctx}}]$ and aim to predict $y_{\text{qy}}$. Trapezoids and triangles are dense layers, with the shape indicating whether the dimension is increased or decreased. After summing the linear embedding of $X$ and $y_{\text{ctx}}$ (shape $(B, N, d)$), we pass them into the transformer model. Depending on whether $y$ is a classification or regression target, we use the appropriate head and either the cross entropy or mean-squared loss, respectively.

thus increasing GPU memory requirements. Instead, we use the approximate retrieval technique presented by Thomas et al. (2024) where local groups of points are sampled and randomly split into a context vector and a query vector, thus allowing context sharing between points while being more efficient. During inference, we perform one $k$NN search per query point, with details in Figure 5.

We retrieve neighbours based on a simple distance ($L_2$ or dot-product) in the normalized original feature space as done by Thomas et al. (2024). There are three main reasons for this: 1) we cannot simply learn good table embeddings – the meaning of values and features is extremely table-dependent, and thus such embeddings would likely need to be learned in-context, defeating the purpose of retrieval in the first place; 2) contrary to applications like RAG (Lewis et al., 2020) where only a few samples are retrieved, we retrieve up to 1,024 samples during training, which increases our probability of sampling at least a few relevant points; and 3) even in LLMs, sparse methods like BM25 that are closer to data space are competitive (Nogueira & Cho, 2019). We show in Figure 4b that this technique improves the performance of our model compared to retrieval limited to inference time.

### 3.4 INFERENCE STRATEGIES

Recall from Section 3.1 that our architecture needs a pre-defined maximum number of features $F_{\max}$ and classes $C_{\max}$. We discuss how to overcome these limitations with inference-time techniques.

**Features** When the number of features in a table exceeds $F_{\max}$, we reduce the dimensionality of the table using Principal Component Analysis (PCA) to $F_{\max}$.

**Classes** If a dataset contains $C$ classes with $C > C_{\max}$, we cannot perform classification in a single forward pass. While we could do binary classification in a one-versus-all fashion, this would require $C$ forward passes which may drastically impact the inference speed of our algorithm; some datasets have hundreds of classes. A much more computationally efficient idea is to write $C$ in base $C_{\max}$ and predict each base-$C_{\max}$ digits separately. This then requires only $\left\lceil \log_{C_{\max}}(C) \right\rceil$ forward passes, which is very efficient. For instance, if we only train using $C_{\max} = 10$ classes and have to predict on $C \leq 100$, this requires at most two forward passes: one for each digit of the class to predict.

## 4 DATA

### 4.1 TRAINING DATA

Our training data was collected from OpenML (Vanschoren et al., 2014) and consists of a wide range of public tabular datasets across numerous domains. To find appropriate datasets, we considered the datasets specified in the Grinsztajn et al. (2022), TabZilla (McElfresh et al., 2023), and AMLB (Gijsbers et al., 2024) benchmarks as well as additional datasets found individually. In total, our training data contained 123 datasets, with a total of 32M rows and 2B cells (individual values within each table). 93 datasets had classification targets, 29 datasets had regression targets, and 1

did not have a default target defined; however, we do generate both classification and regression targets from each dataset with our self-supervised approach during training. The complete list of training datasets is provided in Appendix G.

**Comparison with TabLib and Tabula-8B**  Our training data includes orders of magnitude fewer tables compared to Tabula-8B (Gardner et al., 2024), a tabular model based on LLama 3-8B (Dubey et al., 2024) and trained on data from TabLib (Eggert et al., 2023) that sources tables from GitHub and CommonCrawl. However, in the end, Tabula-8B is trained on 8B tokens mostly from very small tables. We can estimate this to represent between 400M and 8B cells (using the fact that 8 values are encoded into 167 tokens from their Figure 2 due to the verbosity of the encoding mechanism), which is within the same order of magnitude as our training data.

## 4.2 Evaluation Data

For our evaluation, we consider two public benchmarks: CC18 (Bischl et al., 2021) for classification tasks and CTR23 (Fischer et al., 2023) for regression tasks.

CC18 is a curated suite of 72 datasets with classification targets originally sourced from OpenML. These datasets each have between 500 and 100,000 instances, less than 5,000 features, and originate from diverse domains such as finance, biology, games, banking, industrial applications, or natural signals such as vision or sound. Datasets were selected according to curation criteria that included avoiding synthetic data, requiring source information, and removing datasets where a simple algorithm achieved 100% accuracy. CC18 is a widely used benchmark for evaluating tabular learning (Bahri et al., 2022; Hollmann et al., 2023; McElfresh et al., 2023).

CTR23 is a benchmark suite of 35 datasets also curated from OpenML. It follows most of the design choices of CC18 but contains regression rather than classification tasks. In particular, it uses the same restrictions on number of samples and features as CC18, but replaces the accuracy restriction with a requirement that a linear model must not achieve $R^2 = 1$ on the selected datasets.

## 4.3 Contamination Analysis

To ensure that the datasets used for training did not contain any information about the evaluation data, we extracted a range of metadata from each dataset and compared them across all pairs of training and evaluation datasets. This includes: i) dataset names, ii) hashes of dataset files, iii) numbers of columns and rows, iv) target mean and variance, v) mean, variance, skew, and kurtosis of each feature, and vi) coefficients of a univariate linear fit between each feature and the target if available.

To allow for efficient pairwise comparisons between all features in all datasets, we use $k$-d trees (Bentley, 1975) constructed for each dataset that contain the feature statistics. Any pairs of datasets with unusual similarities detected were manually evaluated and removed from training if they were found to be related. Since this procedure is primarily based on automated checks, it can be used in the future to further scale our training data.

## 5 Experiments

In this section, we evaluate TabDPT against tuned baselines on different benchmarks, and then provide a detailed analysis of TabDPT by observing its scaling properties, reporting the runtime, and ablating key components.

## 5.1 Evaluation

**Benchmark Suites**  First we compare our method against tuned, competitive baselines including tree-based methods such as XGBoost (Chen & Guestrin, 2016) and CatBoost (Prokhorenkova et al., 2018), strong deep learning baselines such as TabR (Gorishniy et al., 2024) and MLP-PLR (Gorishniy et al., 2022), as well as $k$NN (Fix, 1985). We further use McElfresh et al. (2023)'s csv file containing the performance of a large number of algorithms per hyperparameter, split, and dataset.[4] We obtain results for XGBoost, CatBoost, LightGBM, and MLP from this file. Our protocol is

---
[4] `https://drive.google.com/drive/folders/1cHisTmruPHDCYVOYnaqvTdybLngMkB8R`

| Algorithm | CC18 | | CTR23 | |
| --- | --- | --- | --- | --- |
| | AUC | Accuracy | Correlation | $R^2$ |
| TabDPT | **0.972** ± [0.971, 0.973] | 0.917 ± [0.915, 0.919] | **0.911** ± [0.908, 0.913] | **0.831** ± [0.826, 0.835] |
| TabR | 0.967 ± [0.965, 0.969] | **0.923** ± [0.920, 0.926] | 0.909 ± [0.905, 0.912] | 0.825 ± [0.818, 0.831] |
| MLP-PLR | 0.967 ± [0.965, 0.968] | 0.914 ± [0.911, 0.917] | 0.907 ± [0.904, 0.910] | 0.827 ± [0.822, 0.832] |
| PFN++ (kNN) | 0.970 ± [0.968, 0.972] | 0.913 ± [0.910, 0.916] | 0.888 ± [0.881, 0.894] | 0.792 ± [0.782, 0.801] |
| XGBoost | 0.966 ± [0.964, 0.967] | 0.911 ± [0.909, 0.913] | 0.904 ± [0.900, 0.907] | 0.820 ± [0.814, 0.825] |
| LightGBM | 0.962 ± [0.960, 0.964] | 0.908 ± [0.906, 0.910] | 0.900 ± [0.896, 0.904] | 0.809 ± [0.803, 0.815] |
| CatBoost | 0.959 ± [0.958, 0.961] | 0.903 ± [0.901, 0.905] | 0.897 ± [0.890, 0.903] | 0.802 ± [0.794, 0.810] |
| TabPFN (kNN) | 0.959 ± [0.955, 0.962] | 0.884 ± [0.881, 0.887] | N/A | N/A |
| TabPFN | 0.939 ± [0.935, 0.943] | 0.852 ± [0.849, 0.855] | N/A | N/A |
| MLP | 0.910 ± [0.907, 0.913] | 0.863 ± [0.860, 0.866] | N/A | N/A |
| kNN | 0.874 ± [0.869, 0.879] | 0.866 ± [0.862, 0.871] | 0.671 ± [0.654, 0.687] | 0.466 ± [0.446, 0.485] |

Table 1: Results on CC18 and CTR23. We report four metrics and their $95\%$ confidence intervals. The best algorithm is bolded for each metric. Furthermore, we underline an algorithm's score if its confidence interval overlaps with the highest score's interval. TabDPT performs strongly across all metrics on both classification and regression, although regression has much higher uncertainty.

the following: if we have access to the hyperparameter optimization (HPO) search from McElfresh et al. (2023), we use those numbers. However, for algorithm and dataset combinations that took 5+ hours to train, the csv entry is missing. For those cases specifically we compute the performance with default hyperparameters.[5] For CTR23, we run a HPO search with search space similar to the TabZilla protocol for XGBoost, CatBoost, and LightGBM, using the code repository from Gorishniy et al. (2024).[6] For TabR, MLP-PLR, and $k$NN, we also use that repository, with the predefined search space and 30 rounds for both CC18 and CTR23.

We choose the best hyperparameters for each dataset fold individually based on the validation performance. In addition, we compare to other ICL baselines including TabPFN (Hollmann et al., 2023), and TabPFN ($k$NN) (Thomas et al., 2024) which retrieves neighbours of each query at inference time. We also introduce PFN++, our improved TabPFN implementation that additionally performs regression. PFN++ has the same architecture and training procedure as TabDPT but it uses the same synthetic data generator as Hollmann et al. (2023) for training. PFN++ ($k$NN) also includes retrieval at test time similar to TabPFN ($k$NN). Details of PFN++ can be found in Appendix I.1.

Finally, we run all methods on at least two different splits of the data and report $95\%$ confidence intervals using bootstrapping on the interquartile mean (IQM) of each metric, following the recommendations of Agarwal et al. (2021). Our model, TabDPT, is a 78M parameter model with 16 transformer layers pre-trained for 600K steps. All model training and inference can be done on a single Nvidia A100 GPU with 40 GB of memory. We observe in Table 1 that TabDPT performs competitively with all the hyperparameter-tuned baselines on both classification and regression, using only forward passes and no further tuning.

**Win-Rate and Comparison with Tabula-8B**  In order to compare with Tabula-8B (Gardner et al., 2024), we gather the results they have on a subset of 61 datasets from CC18 for three models: Tabula-8B with 32-sample context length, Tabula-8B zero shot, and the random baseline. As they only report accuracy, we compute the win-rate for each pair of algorithms by assigning an algorithm a "win" if it achieves a higher accuracy score on CC18 or $R^2$ score on CTR23, for a given dataset and fold. In Figure 3a we show the win-rate matrix for a subset of methods including 32-shot Tabula-8B.

**Elo Scores**  More generally, we propose using Elo ratings (Elo, 1967) to compare algorithms in the tabular domain, where no single gold standard benchmark exists and diverse datasets are commonly used for evaluation. By treating each dataset fold and algorithm pair as a "duel" for a given performance metric, we can assign ratings based on relative performance. This method allows for consistent comparison of algorithms over varying collections of datasets when only a subset of all the pairwise comparisons is available.

---

[5] This only concerns 4 large datasets, namely CIFAR10 for CatBoost, and 3 others for LightGBM.
[6] https://github.com/yandex-research/tabular-dl-tabr

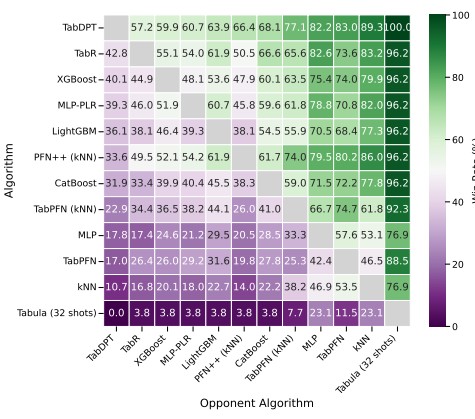 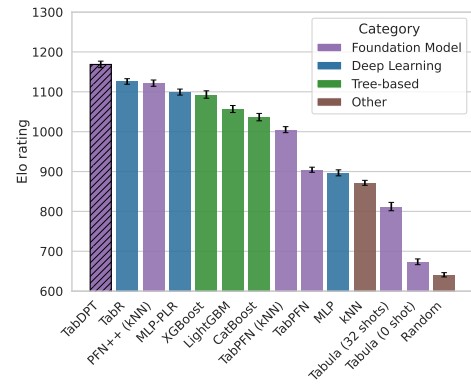

(a) Win-rate matrix for a subset of the methods.   (b) Elo scores (Accuracy, $R^2$) with error bars.

Figure 3: Duel-based metrics computed on accuracy and $R^2$ scores. (a) Win-rate matrix for all datasets available for each algorithm on CC18 and CTR23. (b) Elo ratings: TabDPT, the top-performing algorithm, is highlighted.

We provide the Elo ratings in Figure 3b, which are computed on all the TabZilla scores available to us for the given algorithms. Furthermore, we estimate uncertainty by boostrapping over match order permutations (Boubdir et al., 2023). We also report Glicko2 ratings (Glickman, 2012) in Appendix E as we found Glicko2 to be less sensitive to match order and it computes uncertainty by design, despite Elo being more popular. We include Tabula with 32 shots, 0 shots, and the random baseline. Figure 3b paints a similar picture to the previous results: our method performs best, followed by the strong non-foundation baselines TabR, MLP-PLR, XGBoost, LightGBM, and CatBoost. The LLM-based foundation model, Tabula-8B, is not competitive. Note that for baselines in TabZilla, the missing algorithm-dataset pairs are due to significant time consumption. While one approach would be to treat these missing points as a loss for the given algorithm, we were able to simply omit them due to the paired nature of duel-based metrics. Thus, the outcome is more favourable for the baselines, making the leading position of TabDPT even more impressive.

## 5.2 SCALING LAWS FOR TABULAR DATA

To the best of our knowledge, this work presents the first analysis of scaling laws for tabular foundation models. We observe how performance changes when systematically varying model size by adjusting the number of layers and transformer dimensions, as well as the amount of training data. Our models range from 33K to 78M parameters, trained on data subsets spanning from 52M cells (104K rows) to 2B cells (32M rows). For PFN++, we limit the variation to model size, keeping the prior-generating function fixed. Following Hoffmann et al. (2022), we adopt the joint power-law model $\hat{\ell}(P, D) = A/P^\alpha + B/D^\beta + E$, where $\hat{\ell}$ represents the estimated loss (or another target metric), $P$ denotes the number of parameters, and $D$ the number of tokens, or in our case the number of cells in the entire training set. Notably, although we use row-based encodings, not all rows affect the model (especially the encoder layer) equally, and thus cell count is a better measure for dataset size. We use the improved methodology by Besiroglu et al. (2024) to estimate the parameters $A, B, \alpha, \beta$, and $E$. For the scaling exponents in particular, we find $\alpha = 0.42$ and $\beta = 0.39$, which are within the expected range and are very close to each other, mirroring Hoffmann et al. (2022)'s observation.

In Figure 1, we illustrate the scaling behavior of our models along with the power-law fit. Since we train on both classification and regression tasks, with roughly $50\%$ of the samples in each category, the loss on the $y$-axis represents the average of the cross-entropy loss for classification and $1 - \rho$ for regression, where $\rho$ is the correlation between the prediction and true target, equivalent to MSE for normalized vectors. Note that for visualization purposes we report, on a log-scale, the excess loss $\hat{\ell}(P, D) - E$ (the estimated loss de-biased by $E$) instead of the raw loss. The empirical values $(P, D, \ell(P, D))$ are reported alongside the power-law fit.

We also provide the empirical values for models using the TabPFN prior (Hollmann et al., 2023), shown in green. These models are not fit to the joint power-law model, as the data size $D$ is

not known a priori. However, we can estimate the number of rows or cells seen during training, which totals to approximately 17B rows and 860B cells for all model sizes. As shown in Figure 1, the quality of the data – whether real or synthetic – affects both the shape of the loss curve and the terminal loss. We hypothesize that the synthetic data generated by TabPFN contains many of the "easy" patterns present in real-world data, but not all. This is supported by smaller models outperforming ones trained on real data of up to 2B cells, while larger models trained on TabPFN data perform comparably to models trained on 300-600M real cells.

From Figure 1, we observe that as the models have more parameters, their performance becomes more predictable. However, for larger models on smaller amounts of data, the loss is greater than predicted (and in some cases unstable, see Appendix F), which could indicate signs of overfitting. Additionally, for the joint classification and regression loss we observe a behaviour predictable by power-law models, but neither classification nor regression alone is explained quite as well. In Appendix F, we discuss the scaling analysis in more detail.

Our analysis suggests an important insight: although neural network-based methods have struggled with tabular data for a long time, performance continues to improve as model size and data quantity increase, much like text and image data.

## 5.3 TRAINING AND INFERENCE SPEED

In Figure 4a, we approximate the speed of each algorithm by computing the median time it takes to perform a full training and evaluation – including HPO search – on datasets with size larger than 10,000 rows from CC18. From this, we compute the median time to process 1,000 rows, along with the $25^{th}$ and $75^{th}$ percentiles. We also note, although it is not shown, that average time is usually much higher than median for the baselines. We report the runtimes of TabDPT models with different context sizes, indicated by the number labelling TabDPT points. It is also worth noting that even the biggest TabDPT model with the largest context size is at least one order of magnitude (up to 4) faster than the baseline models. While the baseline models need to train on each dataset separately, TabDPT is much faster thanks to its ICL capability.

In addition, since TabDPT performs inference with a fixed batch and feature size, our speed per 1,000 rows is very consistent. We also add TabDPT (subsampling) which uses a shared random context to classify all the test points – in the style of TabPFN – allowing for even faster inference at the cost of performance. However, while tree-based and DL baselines offer faster inference after training, their efficiency depends on the scenario: TabDPT is advantageous for streaming data requiring frequent retraining, whereas traditional models are preferable for fixed data needing rapid inference.

In the pre-training phase, we have also observed that TabDPT achieves a higher performance than PFN++ within the same number of epochs especially early on during training. In Figure 13, we can see that TabDPT obtains lower test loss given a fixed compute budget, especially for larger models. This clearly highlights the importance of real data compared to synthetic data.

## 5.4 ABLATIONS

In this section, we ablate key components in our training and inference strategies. All models are trained for 700 epochs for a fair comparison. In Figure 4b, we report the reduction in performance by removing key components in the training or inference pipeline; a higher bar indicates a greater reduction in performance. For computational reasons, all comparisons are done against a smaller base model with 28M parameters and 12 layers, and the inference is done using 512 context size with retrieval as done by Thomas et al. (2024).

**Training Ablations** Firstly, we assess the importance of our SSL approach, where columns are randomly sampled as targets during training. To ablate this, we only use the original target during training and we observe the greatest loss in performance overall, as shown under "Supervised Target (Training)" in Figure 4b. This underscores the importance of SSL in our training process. Secondly, using subsampling instead of retrieval during training – but still keeping retrieval during inference – also leads to a performance drop, albeit not as drastic as before.

**Inference Ablations** Similarly to Thomas et al. (2024), we find that using subsampling instead of retrieval during inference decreases performance as indicated in the second column in Figure 4b.

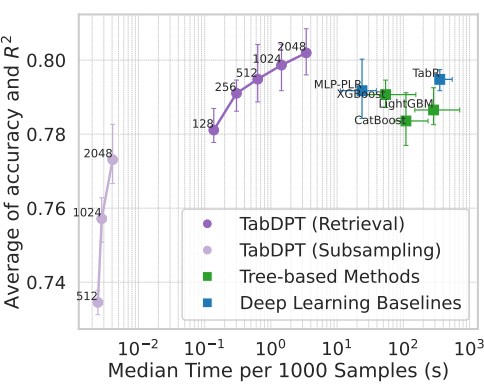

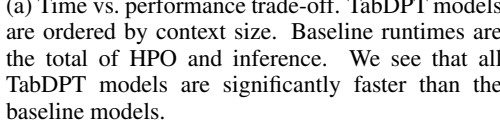

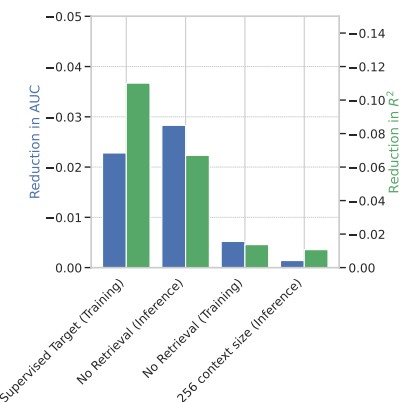

(a) Time vs. performance trade-off. TabDPT models are ordered by context size. Baseline runtimes are the total of HPO and inference. We see that all TabDPT models are significantly faster than the baseline models.

(b) Ablating key components in training and inference. A higher blue bar and a higher green bar indicate greater reduction in AUC and $R^2$, respectively.

Figure 4: Runtime analysis (left) and ablation study (right).

Lastly, using a smaller context size also decreases performance as expected, although it does not decrease nearly as much as the other important components discussed above.

## 6 LIMITATIONS, CONCLUSION, AND FUTURE WORK

While our model achieves very competitive performance on two popular tabular benchmarks, it is still subject to some important limitations that have not been addressed in this work.

i) As of now, the current model cannot use textual information. Note that it is still possible to modify the architecture to use embeddings of feature names, but we did observe overfitting when we attempted it. We believe training the model on more tables with more features overall could lead to improvements but we leave this to future work. ii) We still suffer from the limitation on the number of features, and the non-invariance to the class and feature ordering, inherited from the "attention over rows" architecture of TabPFN (Hollmann et al., 2023). The ordering dependence is mitigated through randomization during training and we proposed inference techniques to address the former limitations, although a better outcome would be finding a performant architecture that inherently does not suffer from these limitations. iii) We have made some assumptions on the types of data we handle: these are rectangular (i.e., not hierarchical or nested) tables which are i.i.d. (i.e., having no time component or distribution shift between training data and test/inference data). iv) Unlike foundation models in other domains, we have not shown generative (Loaiza-Ganem et al., 2024) capabilities. We anticipate that TabDPT could be complemented with ideas from, e.g., (Ma et al., 2023), (Ma et al., 2024) or (van Breugel et al., 2024). v) Even though our runtime on new tasks is extremely fast compared to the baseline models, the pre-training of TabDPT is still itself time- and resource-consuming, although we will provide the weights of TabDPT to help mitigate this. Furthermore, there may be specific applications or settings where TabDPT and other ICL-based techniques require additional fine-tuning, which may make them slower in comparison.

While deep learning has traditionally struggled with tabular data, our findings demonstrate that, with the appropriate architecture and data utilization, similar scaling laws to those seen in other domains can emerge. This suggests that tabular data is not inherently unique, and we anticipate that larger tabular models trained on increasing amounts of data will become the norm. In this paper, we present TabDPT as a scalable model that achieves strong performance on the CC18 and CTR23 benchmarks, but we expect the field to see the development of even larger models trained on more expansive datasets in the future.

REPRODUCIBILITY STATEMENT

We provide a description of the architectural details, hyperparameters and datasets used for both evaluation and training. We intend to release the model weights, the full training and inference code, and some data management functions such as automated tests for contamination.

ETHICS STATEMENT

We do not foresee any ethical concerns with the present research. The creation of a tabular foundation model and studying the scaling of such models are unlikely to be used for harmful purposes. Nevertheless, we do not promote the use of these models for harmful practices.

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

## A  BITTER LESSONS

The architecture of TabDPT is based on TabPFN with some minor modifications (see Appendix H.1). Therefore, it inherits the limitations of TabPFN. To mitigate these, we introduce inference time techniques (see Section 3.4), although we also attempted to overcome these shortcomings during training. The following list contains some of the ideas that either hurt the performance or did not lead to significant improvements. We include this list to emphasize *The Bitter Lesson*[7] and the fact that efficient use of computation and access to high-quality data are the more important factors in driving performance. The list is as follows:

- Different pre-processing techniques that were more robust to outliers, or variants of soft clipping, resulted in no improvement. More advanced methods, such as Robust Scaler and Power Transform, only ended up slowing the training process.

- Class embeddings (either through a separate network or by using class "tokens" in the transformer layer) and computing various similarity metrics between query and class embeddings in a proto-network manner, with the aim of adapting to any number of classes, hurt the performance, especially on real data.

- Different embeddings for $y_{ctx}$, including a dense layer for regression and a dictionary of $C_{max} \times d$ embeddings, with the rationale of informing the model about the task, did not lead to performance improvements in large models with sufficient data.

- Specialized tokens for NaN encoding did not improve performance compared to replacing NaNs with mean values (which are zero after preprocessing). Additionally, appending binary features to differentiate actual zeros from NaNs (indicating that the cell was replaced), effectively doubling the number of features, also failed to improve performance.

- Architectures encoding cells as "tokens", with vertical and horizontal attention, similar to spatial and temporal attention in videos, proved more memory intensive. While equivariance to feature order is desirable, processing tensors of size $(B, N, f, d)$ – where $B$ is batch size, $N$ is the number of rows, $f$ the number of features, and $d$ the embedding dimension – uses much more memory. The simpler architecture with tensors of size $(B, N, d)$ permits a higher embedding dimension $d$.

## B  TRAINING AND INFERENCE DETAILS

In this section we provide some additional details on some important training decisions.

**Preprocessing**  We chose to be very simple and general. All columns containing non-numerical values are mapped to integers using `scikit-learn`'s (Pedregosa et al., 2011) `LabelEncoder` function. The table is then standardized to $0$ mean and unit variance, and outliers beyond $10$ are clipped.

After retrieval, we obtain a local context $X_{ctx}$ and their labels $y_{ctx}$. We make sure to standardize the context before the forward pass of our model to avoid distribution shifts and also standardize $y_{ctx}$ if it is a regression target for the same reason.

**Retrieval**  We use the `faiss` library[8] for fast retrieval. All retrievals are done using the raw data space after preprocessing, as in Thomas et al. (2024).

**Missing Value Encoding**  We tried several strategies, including concatenating our features with a mask indicating whether each value was originally a missing value or not, however we never saw any performance gain from it. In the end, we simply zero out the missing values and let the model learn to be robust to potentially inaccurate values in the input. Note that zeroing out is done post normalization, meaning missing values are replaced with the mean.

**Optimizer:**  We use the Schedule Free optimizer from Defazio et al. (2024) with AdamW (Loshchilov & Hutter, 2019). We observed significant increase in performance and optimization speed compared to a cosine scheduler.

---

[7] http://www.incompleteideas.net/IncIdeas/BitterLesson.html
[8] https://github.com/facebookresearch/faiss

**Regularization** By default, the training of the model can lead to gradient explosion. We have found it critical to regularize the model. While using label smoothing is helpful, we found that increasing the amount of weight decay was key to mitigate instabilities.

## C PSEUDO-CODE ALGORITHMS

In this section, we show (semi-)pseudo-code blocks of components needed for training for our model. Firstly, in Code Block 1, we show the `PyTorch Dataloader` component. In the initialization phase, we first process the downloaded data and features by filling in missing values with the mean column values and create a `faiss` index for fast retrieval later on. Next. in each worker within the `getitem()` function, we first sample a random dataset, then we sample a random query within the dataset. After that, we mask out the target column and retrieve its approximate neighbours. Then we process the features and targets by random sub-sampling and random partitioning.

Moreover, in Code Block 2, within each training step, we partition both the data X and targets y into context and query points by sampling an integer uniformly from 10 to its total length (inclusive of start point but exclusive of endpoint). We call this random evaluation position `eval_pos` in the code block. The points to the left of the evaluation position are then considered context (i.e., `y_ctx`) and the points to the right of the evaluation position are considered queries (i.e., `y_qy`). Finally we calculate the appropriate loss depending on the task and optimize the network.

Code Block 1: Pytorch Dataloader

```python
from torch.utils.data import Dataset
import numpy as np
import random

class TrainingDataset(Dataset):
    def __init__(self, dataset_ids):
        self.datasets = []
        for dataset_id in dataset_ids:
            X <- download dataset using dataset_id
            X <- process features of X (handle missing values, scale)
            knn_index <- compute knn index using FAISS
            self.dataset.append([X, knn_index])

    # Random column subsample and shuffling
    def create_random_columns(self, X):
        N, F = X.shape
        num_features_sampled = random.randint(F // 2, F)
        random_features_indices = np.random.choice(F,
            num_features_sampled, replace=False)
        return X[:, random_features_indices]

    # Generate a random classification or regression target for training
    def generate_random_target(self, y, cls_threshold=10):
        if len(np.unique(y)) > cls_threshold:
            # if there are more than 10 unique values in the target, we
                keep it as regression 70% of the time
            if np.random.rand() > 0.3:
                return y, "regression"
            else:
                # sample a random number of classes by binning and divide
                    into classes
                num_class = np.random.randint(2, cls_threshold)
                cls_boundary = np.random.choice(sorted(np.unique(y))
                    [1:-1], num_class-1, replace=False)
                y = (y[:, None] > cls_boundary[None, :]).sum(1)
                y <- label encode, shuffle y
                return y, "classification"
        else:
            assert len(np.unique(y)) > 1
            y <- label encode, shuffle y
```

```
37              return y, "classification"
38
39      # Generate a sample for retrieval
40      def __getitem__(_):
41          # sample a random dataset
42          sample_id = np.random.choice(len(self.dataset), 1)[0]
43          X_sample, knn_index_sample = self.dataset[sample_id]
44          N, F = X_sample.shape
45
46          # sample a random query from the dataset
47          x_q = X_sample[random.randint(0, N-1)].copy()
48
49          # sample a random column to be the target
50          target_idx = random.randint(0, F-1)
51
52          # retrieve approximate neighbours using x_q with target_idx
                masked
53          x_q[:, target_idx] = 0
54          X_nn <- find k neighbours using knn_index_sample with x_q as
                query
55          y_nn = X_nn[:, target_idx]
56          X_nn = np.delete(X_nn, target_idx, axis=1)
57
58          # subsample and shuffle features
59          X_nn = self.create_random_columns(X_nn)
60
61          # generate random target and task
62          y_nn, task = self.generate_random_target(y_nn)
63
64          return X, y, task
```

Code Block 2: Training Loop

```
1
2   model = Transformer()
3   optimizer = schedulerfree.AdamWScheduleFree()
4
5   for epoch in range(num_epochs):
6     model.train()
7     for X, y, task in train_loader:
8         eval_pos = random.randint(10, len(y))
9         y_ctx, y_qy = y[:eval_pos], y[eval_pos:]
10        y_ctx = zero_pad(y_ctx, N_qy, dim=1)
11
12        output = model(torch.cat(X, y_ctx))
13
14        if task == "classification":
15            loss = cross_entropy_loss(output, y_qy)
16        elif task == "regression":
17            loss = mse_loss(ouput, y_qy)
18
19        opitmizer.zero_grad()
20        loss.backward()
21        optimizer.step()
```

## D  DETAILS ON THE RETRIEVAL

During training, it is not necessary to predict the outcome for only a single point, as the point is not provided in advance. Instead, when multiple points belong to the same neighbourhood, a single model call with shared context can perform prediction for all of them, optimizing both memory and compute during training. Selecting a "local neighbourhood" of points and distributing them between context and query achieves such an arrangement. Specifically, we begin with one point per

dataset ($B$ points in the batch), each with $N$ neighbours. We obtain the input $X$ by either padding or dimensionality reduction, resulting in a shape of $(B, N, F_{\max})$. After constructing the random target $y$ of shape $(B, N)$, we randomly split these into context and query sets along the row dimension $N$.

The goal is to evaluate the loss on $B \times N_{\text{qy}}$ points at once. If exact retrieval were performed, only one prediction per neighbourhood would be possible. This method creates a context that is not only local to each point but also shared across $N_{\text{qy}}$ points. This approach was introduced in Thomas et al. (2024), and more details can be found there. During inference, exact retrieval is required as the query points are fixed. We illustrate the treatment of samples during training and retrieval in Figure 5.

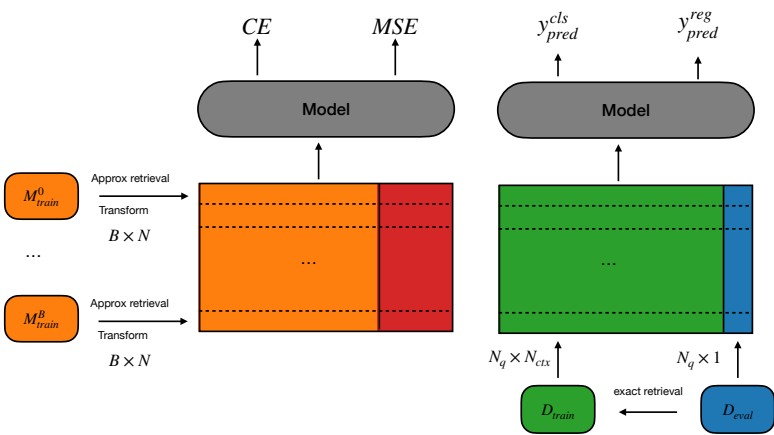

Figure 5: Contrasting the shared context and multiple datasets during training and instance specific context for multiple points in one dataset at inference time. Details are explained in Appendix D.

## E   GLICKO2 RATINGS

Similar to the Elo score in Figure 3b, we plot Glicko2 ratings in Figure 6. The implications from this figure are the same as the ones in Section 5.1.

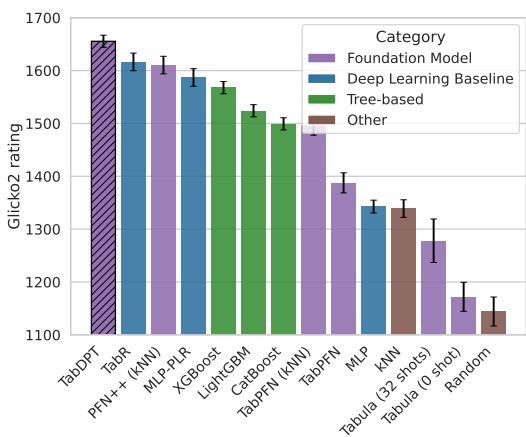

Figure 6: Glicko2 scores (Accuracy, $R^2$) with error bars.

## F   SCALING

As some models can exhibit unstable training characterized by gradient explosion, which is especially true for larger models trained on little data, we filter out checkpoints with a training gradient

norm above $0.9$. We then report the performance, model size, and data size for all our models, provided they have been trained for at least 400 epochs, up to 512 epochs (the predefined maximum epoch parameter for most models). We report several metrics in this section. Metrics averaging classification and regression (in the same way our loss is defined) exhibit scaling laws with similar scaling exponents. Because of the number of models and increasing computational cost, we reduce in-context examples from 1024 to 512 to keep the experiment manageable.

For classification, cross-entropy saturates with larger models and datasets, suggesting the model is close to the optimal loss for its size. In contrast, regression performance continues to improve, especially with more data. We hypothesize this is because regression targets are less randomized compared to classification, where class ordering and membership are shuffled, increasing the classification dataset's effective size. As a result, saturating regression performance for a given model size requires even larger datasets.

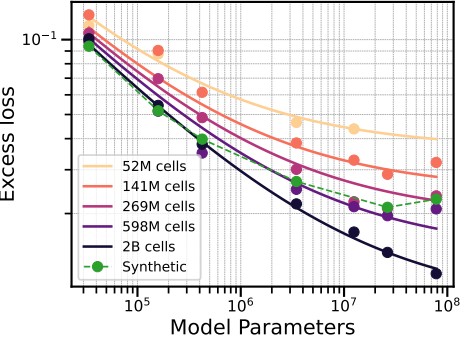
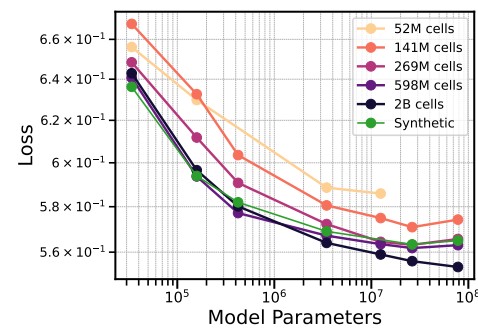

(a) Power-law fit for the average of cross entropy and $1 - \text{correlation}$.

(b) Raw points (no offset) for the average of cross entropy and $1 - \text{correlation}$.

Figure 7: Comparison between the power-law fit and raw points.

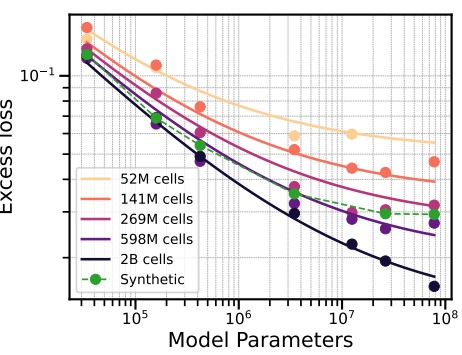
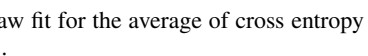
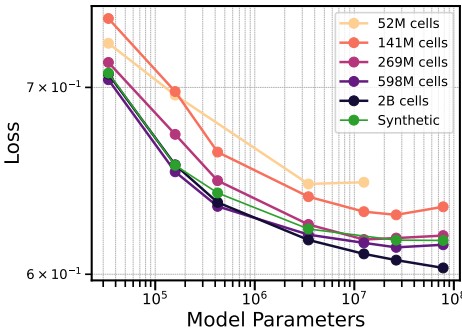

(a) Power-law fit for the average of cross entropy and $1 - R^2$.

(b) Raw points (no offset) for the average of cross entropy and $1 - R^2$.

Figure 8: Comparison between the power-law fit and raw points.

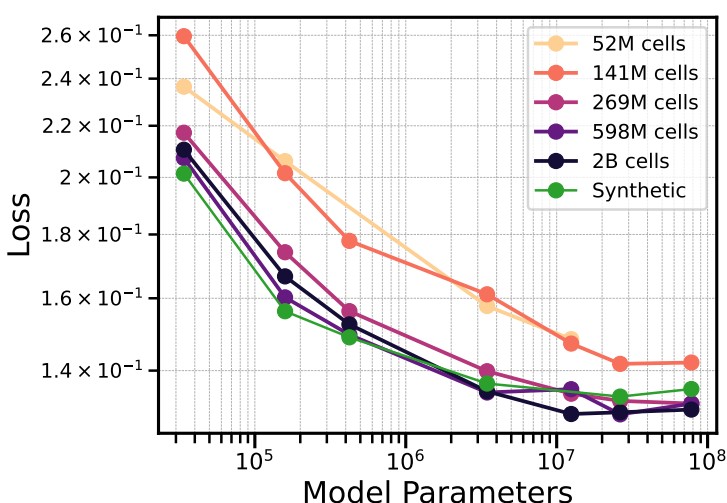

Figure 9: Raw points for $1 - \text{accuracy}$ with original y-axis.

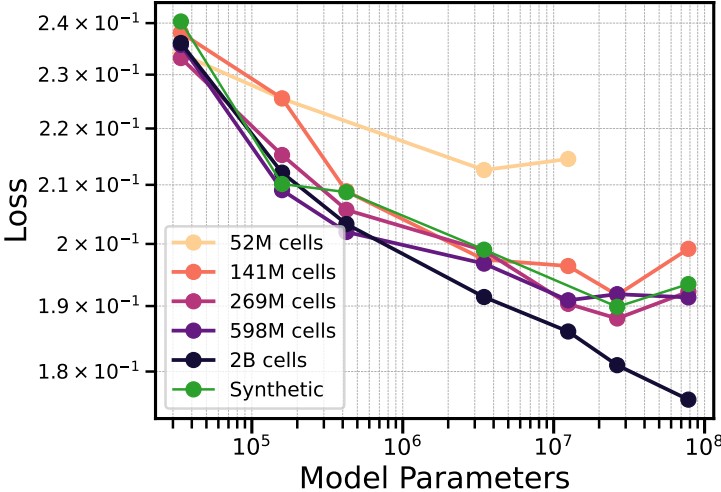

Figure 10: Raw points for $1 - \text{correlation}$ with original y-axis.

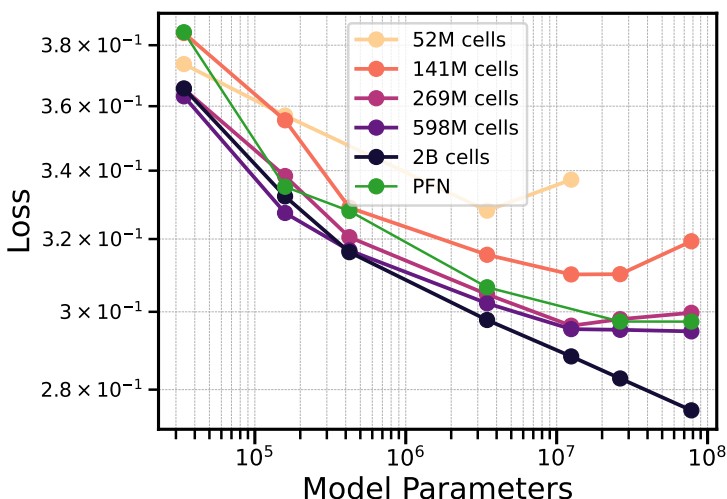

Figure 11: Raw points for $1 - R^2$ with original y-axis.

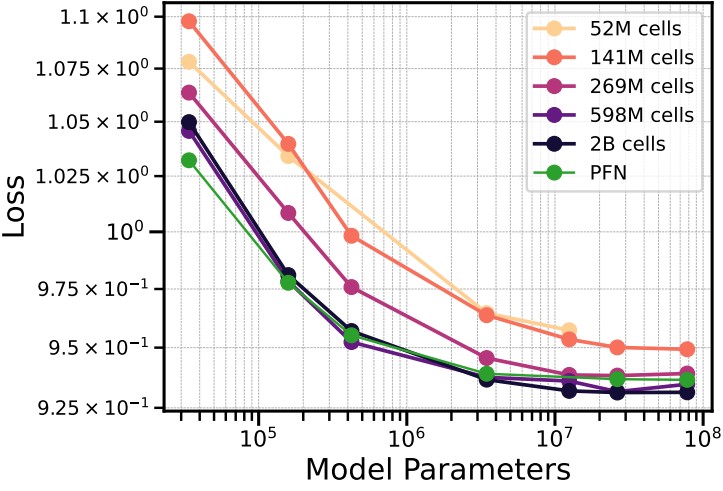

Figure 12: Raw points for cross entropy with original y-axis.

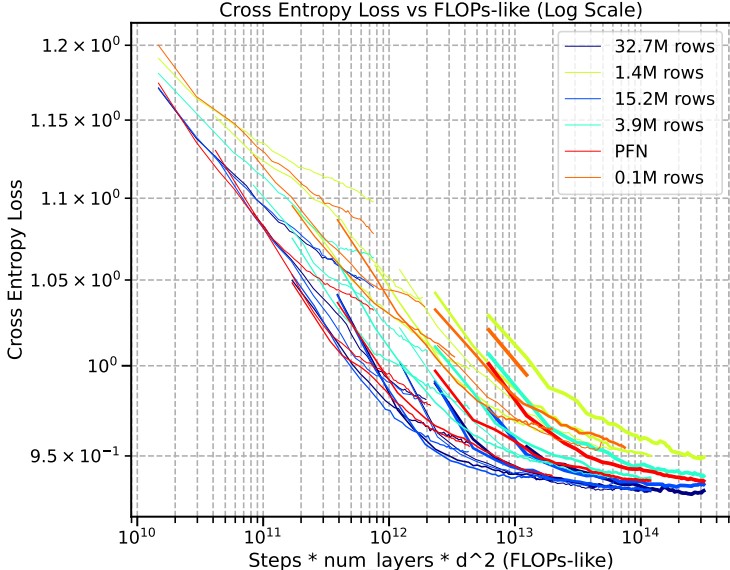

Figure 13: Cross entropy loss for all models sizes and data sizes vs. compute units.

## G TRAINING DATASETS

Figure 14 provides an overview of the sizes and domains of the training datasets and Table 2 provides a full list of the datasets.

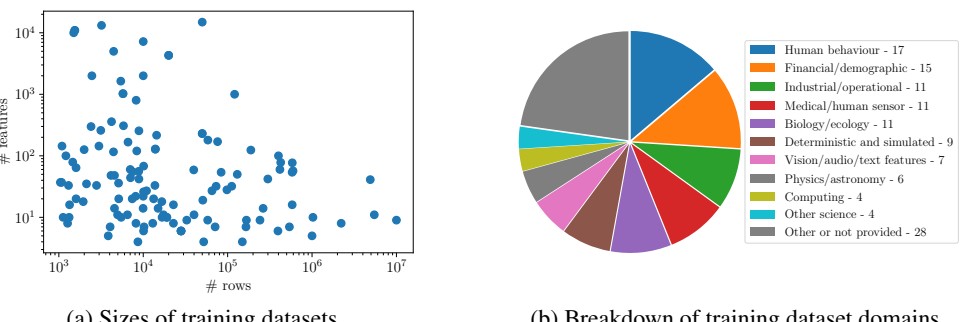

(a) Sizes of training datasets.

(b) Breakdown of training dataset domains.

Figure 14: Breakdown of training datasets in terms of sizes and domains.

Table 2: Details for all training datasets: OpenML Dataset ID, name, dimensions (rows, features, cells), percent of missing cells, target type (classification/regression), domain.

| OpenML Dataset ID | Name | # rows | # feat. | # cells | % miss. | Target type | Domain |
|---|---|---|---|---|---|---|---|
| 24 | mushroom | 8124 | 22 | 187K | 1.4 | Class. | Biology/ecology |
| 30 | page-blocks | 5473 | 10 | 60K | 0.0 | Class. | Vision/audio/text features |
| 184 | kropt | 28056 | 6 | 196K | 0.0 | Class. | Deterministic and simulated |
| 273 | IMDB.drama | 120919 | 1001 | 121M | 0.0 | Class. | Other or not provided |
| 312 | scene | 2407 | 299 | 722K | 0.0 | Class. | Vision/audio/text features |
| 375 | JapaneseVowels | 9961 | 14 | 149K | 0.0 | Class. | Vision/audio/text features |
| 382 | ipums_la_97-small | 7019 | 60 | 428K | 11.4 | Class. | Financial/demographic |
| 389 | fbis.wc | 2463 | 2000 | 4.9M | 0.0 | Class. | Vision/audio/text features |
| 396 | la1s.wc | 3204 | 13195 | 42M | 0.0 | Class. | Vision/audio/text features |
| 802 | pbcseq | 1945 | 18 | 37K | 3.2 | Class. | Medical/human sensor |
| 816 | puma8NH | 8192 | 8 | 74K | 0.0 | Class. | Deterministic and simulated |
| 821 | house_16H | 22784 | 16 | 387K | 0.0 | Class. | Financial/demographic |
| 843 | house_8L | 22784 | 8 | 205K | 0.0 | Class. | Financial/demographic |
| 846 | elevators | 16599 | 18 | 315K | 0.0 | Class. | Other or not provided |

| OpenML Dataset ID | Name | # rows | # feat. | # cells | % miss. | Target type | Domain |
|---|---|---|---|---|---|---|---|
| 871 | pollen | 3848 | 5 | 23K | 0.0 | Class. | Biology/ecology |
| 930 | colleges_usnews | 1302 | 33 | 44K | 18.2 | Class. | Other or not provided |
| 966 | analcatdata_halloffame | 1340 | 16 | 23K | 0.1 | Class. | Other or not provided |
| 981 | kdd_internet_usage | 10108 | 68 | 697K | 0.4 | Class. | Financial/demographic |
| 1002 | ipums_la_98-small | 7485 | 55 | 419K | 7.9 | Class. | Financial/demographic |
| 1018 | ipums_la_99-small | 8844 | 56 | 504K | 7.0 | Class. | Financial/demographic |
| 1036 | sylva_agnostic | 14395 | 216 | 3.1M | 0.0 | Class. | Biology/ecology |
| 1037 | ada_prior | 4562 | 14 | 68K | 0.1 | Class. | Financial/demographic |
| 1043 | ada_agnostic | 4562 | 48 | 224K | 0.0 | Class. | Financial/demographic |
| 1044 | eye_movements | 10936 | 27 | 306K | 0.0 | Class. | Medical/human sensor |
| 1111 | KDDCup09_appetency | 50000 | 230 | 12M | 61.9 | Class. | Human behaviour |
| 1112 | KDDCup09_churn | 50000 | 230 | 12M | 61.9 | Class. | Industrial/operational |
| 1116 | musk | 6598 | 167 | 1.1M | 0.0 | Class. | Other science |
| 1118 | chess | 28056 | 6 | 196K | 0.0 | Class. | Deterministic and simulated |
| 1120 | MagicTelescope | 19020 | 10 | 209K | 0.0 | Class. | Physics/astronomy |
| 1130 | OVA_Lung | 1545 | 10935 | 17M | 0.0 | Class. | Biology/ecology |
| 1142 | OVA_Endometrium | 1545 | 10935 | 17M | 0.0 | Class. | Biology/ecology |
| 1169 | airlines | 539383 | 7 | 4.3M | 0.0 | Class. | Industrial/operational |
| 1444 | PizzaCutter3 | 1043 | 37 | 40K | 0.0 | Class. | Other or not provided |
| 1453 | PieChart3 | 1077 | 37 | 41K | 0.0 | Class. | Other or not provided |
| 1457 | amazon-commerce-reviews | 1500 | 10000 | 15M | 0.0 | Class. | Vision/audio/text features |
| 1459 | artificial-characters | 10218 | 7 | 82K | 0.0 | Class. | Deterministic and simulated |
| 1466 | cardiotocography | 2126 | 35 | 77K | 0.0 | Class. | Medical/human sensor |
| 1471 | eeg-eye-state | 14980 | 14 | 225K | 0.0 | Class. | Medical/human sensor |
| 1476 | gas-drift | 13910 | 128 | 1.8M | 0.0 | Class. | Other science |
| 1477 | gas-drift-different-concentrations | 13910 | 129 | 1.8M | 0.0 | Class. | Other science |
| 1479 | hill-valley | 1212 | 100 | 122K | 0.0 | Class. | Deterministic and simulated |
| 1481 | kr-vs-k | 28056 | 6 | 196K | 0.0 | Class. | Deterministic and simulated |
| 1483 | ldpa | 164860 | 7 | 1.3M | 0.0 | Class. | Medical/human sensor |
| 1493 | one-hundred-plants-texture | 1599 | 64 | 104K | 0.0 | Class. | Biology/ecology |
| 1503 | spoken-arabic-digit | 263256 | 14 | 3.9M | 0.0 | Class. | Vision/audio/text features |
| 1507 | twonorm | 7400 | 20 | 155K | 0.0 | Class. | Deterministic and simulated |
| 1509 | walking-activity | 149332 | 4 | 747K | 0.0 | Class. | Medical/human sensor |
| 1567 | poker-hand | 1025009 | 10 | 11M | 0.0 | Class. | Deterministic and simulated |
| 1568 | nursery | 12958 | 8 | 117K | 0.0 | Class. | Financial/demographic |
| 1596 | covertype | 581012 | 54 | 32M | 0.0 | Class. | Biology/ecology |
| 3050 | QSAR-TID-11 | 5742 | 1024 | 5.9M | 0.0 | Reg. | Medical/human sensor |
| 3277 | QSAR-TID-10980 | 5766 | 1024 | 5.9M | 0.0 | Reg. | Medical/human sensor |
| 4135 | Amazon_employee_access | 32769 | 9 | 328K | 0.0 | Class. | Industrial/operational |
| 4535 | Census-Income | 299285 | 42 | 13M | 0.0 | None | Financial/demographic |
| 4549 | Buzzinsocialmedia_Twitter | 583250 | 77 | 45M | 0.0 | Reg. | Human behaviour |
| 23380 | cjs | 2796 | 33 | 95K | 73.8 | Class. | Biology/ecology |
| 23512 | higgs | 98050 | 28 | 2.8M | 0.0 | Class. | Physics/astronomy |
| 40536 | SpeedDating | 8378 | 120 | 1.0M | 1.8 | Class. | Human behaviour |
| 40646 | GAMETES_Epistasis_2-Way_20atts_0.1H_EDM-1_1 | 1600 | 20 | 34K | 0.0 | Class. | Biology/ecology |
| 40679 | magic | 19020 | 10 | 209K | 0.0 | Class. | Physics/astronomy |
| 40680 | mofn-3-7-10 | 1324 | 10 | 15K | 0.0 | Class. | Other or not provided |
| 40685 | shuttle | 58000 | 9 | 580K | 0.0 | Class. | Physics/astronomy |
| 40706 | parity5_plus_5 | 1124 | 10 | 12K | 0.0 | Class. | Deterministic and simulated |
| 40733 | yeast | 1269 | 8 | 11K | 0.0 | Class. | Biology/ecology |
| 40900 | Satellite | 5100 | 36 | 189K | 0.0 | Class. | Physics/astronomy |
| 41138 | APSFailure | 76000 | 170 | 13M | 8.3 | Class. | Industrial/operational |
| 41142 | christine | 5418 | 1636 | 8.9M | 0.0 | Class. | Other or not provided |
| 41143 | jasmine | 2984 | 144 | 433K | 0.0 | Class. | Other or not provided |
| 41144 | madeline | 3140 | 259 | 816K | 0.0 | Class. | Other or not provided |
| 41145 | philippine | 5832 | 308 | 1.8M | 0.0 | Class. | Other or not provided |
| 41146 | sylvine | 5124 | 20 | 108K | 0.0 | Class. | Other or not provided |
| 41147 | albert | 425240 | 78 | 34M | 8.2 | Class. | Other or not provided |
| 41150 | MiniBooNE | 130064 | 50 | 6.6M | 0.0 | Class. | Physics/astronomy |
| 41156 | ada | 4147 | 48 | 203K | 0.0 | Class. | Other or not provided |
| 41159 | guillermo | 20000 | 4296 | 86M | 0.0 | Class. | Other or not provided |
| 41161 | riccardo | 20000 | 4296 | 86M | 0.0 | Class. | Other or not provided |
| 41162 | kick | 72983 | 32 | 2.4M | 6.4 | Class. | Industrial/operational |
| 41163 | dilbert | 10000 | 2000 | 20M | 0.0 | Class. | Other or not provided |
| 41164 | fabert | 8237 | 800 | 6.6M | 0.0 | Class. | Other or not provided |
| 41165 | robert | 10000 | 7200 | 72M | 0.0 | Class. | Other or not provided |
| 41166 | volkert | 58310 | 180 | 11M | 0.0 | Class. | Other or not provided |
| 41167 | dionis | 416188 | 60 | 25M | 0.0 | Class. | Other or not provided |
| 41168 | jannis | 83733 | 54 | 4.6M | 0.0 | Class. | Other or not provided |
| 41169 | helena | 65196 | 27 | 1.8M | 0.0 | Class. | Other or not provided |
| 41434 | Click_prediction_small | 39948 | 11 | 479K | 0.0 | Class. | Human behaviour |
| 41540 | black_friday | 166821 | 9 | 1.7M | 0.0 | Reg. | Human behaviour |
| 41980 | SAT11-HAND-runtime-Reg. | 4440 | 116 | 519K | 5.3 | Reg. | Computing |
| 42563 | house_prices_nominal | 1460 | 79 | 117K | 6.0 | Reg. | Financial/demographic |
| 42572 | Santander_transaction_value | 4459 | 4991 | 22M | 0.0 | Reg. | Human behaviour |
| 42705 | Yolanda | 400000 | 100 | 40M | 0.0 | Reg. | Other or not provided |

| OpenML Dataset ID | Name | # rows | # feat. | # cells | % miss. | Target type | Domain |
|---|---|---|---|---|---|---|---|
| 42724 | OnlineNewsPopularity | 39644 | 59 | 2.4M | 0.0 | Reg. | Human behaviour |
| 42727 | colleges | 7063 | 44 | 318K | 33.5 | Reg. | Other or not provided |
| 42728 | Airlines_DepDelay_10M | 10000000 | 9 | 100M | 0.0 | Reg. | Industrial/operational |
| 42730 | us_crime | 1994 | 126 | 253K | 15.6 | Reg. | Financial/demographic |
| 42732 | sf-police-incidents | 2215023 | 8 | 20M | 0.0 | Class. | Human behaviour |
| 42734 | okcupid-stem | 50789 | 19 | 1.0M | 16.0 | Class. | Human behaviour |
| 42742 | porto-seguro | 595212 | 57 | 35M | 2.5 | Class. | Human behaviour |
| 42746 | KDDCup99 | 4898431 | 41 | 206M | 0.0 | Class. | Computing |
| 43071 | MIP-2016-Reg. | 1090 | 144 | 158K | 0.0 | Reg. | Computing |
| 43072 | KDDCup09-Upselling | 50000 | 14891 | 745M | 2.6 | Class. | Human behaviour |
| 44055 | analcatdata_supreme | 4052 | 7 | 32K | 0.0 | Reg. | Other or not provided |
| 44056 | visualizing_soil | 8641 | 4 | 43K | 0.0 | Reg. | Biology/ecology |
| 44061 | Mercedes_Benz_Greener_Manufacturing | 4209 | 359 | 1.5M | 0.0 | Reg. | Industrial/operational |
| 44063 | Bike_Sharing_Demand | 17379 | 11 | 209K | 0.0 | Reg. | Human behaviour |
| 44065 | nyc-taxi-green-dec-2016 | 581835 | 16 | 9.9M | 0.0 | Reg. | Human behaviour |
| 44068 | particulate-matter-ukair-2017 | 394299 | 6 | 2.8M | 0.0 | Reg. | Other or not provided |
| 44069 | SGEMM_GPU_kernel_performance | 241600 | 9 | 2.4M | 0.0 | Reg. | Computing |
| 44089 | credit | 16714 | 10 | 184K | 0.0 | Class. | Financial/demographic |
| 44122 | pol | 10082 | 26 | 272K | 0.0 | Class. | Industrial/operational |
| 44136 | wine_quality | 6497 | 11 | 78K | 0.0 | Reg. | Human behaviour |
| 44137 | Ailerons | 13750 | 33 | 468K | 0.0 | Reg. | Other or not provided |
| 44145 | sulfur | 10081 | 6 | 71K | 0.0 | Reg. | Other science |
| 45020 | default-of-credit-card-clients | 13272 | 20 | 279K | 0.0 | Class. | Financial/demographic |
| 45022 | Diabetes130US | 71090 | 7 | 569K | 0.0 | Class. | Medical/human sensor |
| 45026 | heloc | 10000 | 22 | 230K | 0.0 | Class. | Financial/demographic |
| 45032 | yprop_4_1 | 8885 | 42 | 382K | 0.0 | Reg. | Medical/human sensor |
| 45038 | road-safety | 111762 | 32 | 3.7M | 0.0 | Class. | Human behaviour |
| 45039 | compas-two-years | 4966 | 11 | 60K | 0.0 | Class. | Human behaviour |
| 45041 | topo_2_1 | 8885 | 255 | 2.3M | 0.0 | Reg. | Medical/human sensor |
| 45043 | seattlecrime6 | 52031 | 4 | 260K | 0.0 | Reg. | Human behaviour |
| 45045 | delays_zurich_transport | 5465575 | 11 | 66M | 0.0 | Reg. | Industrial/operational |
| 45046 | Allstate_Claims_Severity | 188318 | 124 | 24M | 0.0 | Reg. | Industrial/operational |
| 45047 | Airlines_DepDelay_1M | 1000000 | 5 | 6.0M | 0.0 | Reg. | Industrial/operational |

# H  MODEL ARCHITECTURE AND HYPERPARAMETERS

## H.1  ARCHITECTURE DETAILS

The model architecture comprises multiple transformer encoder layers, an input encoder, and task-specific output heads. The key architectural parameters are summarized in Tables 4 and 5.

Table 4: Architectural Parameters

| Parameter | Value |
|---|---|
| Number of Attention Heads | 4 |
| Feedforward Network Factor | 2 |
| Maximum Number of Classes | 10 |
| Maximum Number of Features | 100 |
| Normalization First | Yes |
| Dropout Rate | 0.0 |

Table 5: Number of Layers and Transformer Dimensions

| Number of Layers | Transformer Dimension |
|---|---|
| 3 | 32 |
| 4 | 64 |
| 5 | 96 |
| 6 | 256 |
| 10 | 384 |
| 12 | 512 |
| 16 | 768 |

**Main Differences with TabPFN's Architecture:** Our backbone is built on the TabPFN one. The most salient changes are: 1) using pre-norm transformer layers, 2) using an RMS normalization layer for the input. TabPFN had to normalize by the number of original features before padding, while this layer eliminates the need, and 3) the two output heads. While we have tried many architectural changes, the simplest choices ended up being the most scalable (see Appendix A). TabPFN also uses 12 layers and $d = 512$ which is the second-largest size we tried.

## H.2 COMPONENT OVERVIEW

- **Input Encoder**: Projects input features to the transformer input dimension.
- **Transformer Encoder**: Consists of multiple layers with specified attention heads and feed-forward dimensions based on the number of layers.
- **Output Heads**: Separate heads for classification and regression tasks.

## H.3 TRAINING PROCEDURE

- Schedule-free optimizer (Defazio et al., 2024) with a learning rate of $5 \times 10^{-4}$ and weight decay of $5 \times 10^{-2}$ is used for training.
- Batch size is 256.
- Model parameters are in brain float 16-bit (`bfloat16`) format.
- Label smoothing is applied with a factor of $0.1$.
- Total length of query and context during training is 1024.

## I ADDITIONAL EXPERIMENTS

### I.1 DETAILS OF PFN++

**PFN++** In addition to our main model TabDPT, we also introduce PFN++, which is an improved version of the original TabPFN (Hollmann et al., 2023). PFN++ uses the same prior generator as Hollmann et al. (2023) but it shares the same model architecture and training procedure as Tab-DPT, detailed in Appendix H.

Moreover, unlike TabPFN, PFN++ can also perform regression. In addition to TabPFN's classification targets, we create synthetic regression targets for training PFN++ during the prior fitting stage. In the TabPFN implementation, targets are first sampled from a Structural Causal Model (SCM), then they are binned and transformed into classification targets. We slightly modify this method for regression purposes by taking the raw outputs from the SCM and normalizing them using the Z-score.

**TabPFN** We use the officially released checkpoint of TabPFN[9] for our experiments. In Hollmann et al. (2023), the TabPFN model is optionally ensembled by randomly shuffling features and classes. We omit feature or class ensembling in all of our experiments across all models to ensure fair comparisons.

**PFN++ vs. TabPFN** We compare the performance of PFN++ and the original TabPFN in Table 6. We use the same 28M parameter model for both methods. The experimental results are obtained using the 30 datasets from CC18 used by Hollmann et al. (2023). We experiment with 2 different folds defined by McElfresh et al. (2023). The final results reported for AUC and Accuracy are averaged over 2 folds and 30 datasets. The results show that PFN++ outperforms TabPFN on both metrics.

| Algorithm | AUC | Accuracy |
|-----------|--------|----------|
| TabPFN | 0.8939 | 0.8262 |
| PFN++ | **0.9063** | **0.8421** |

Table 6: Average AUC and accuracy for TabPFN and PFN++ on 30 selected datasets used by Hollmann et al. (2023). PFN++ outperforms TabPFN on both metrics.

---

[9] https://github.com/automl/TabPFN/blob/main/tabpfn/models_diff/prior_diff_real_checkpoint_n_0_epoch_42.cpkt

## I.2 ADDITIONAL RESULTS ON CC18 AND CTR23

We report here the results for the mean estimator using bootstrapping. Note that the confidence intervals are larger as this estimator is less robust to outliers. This is notably the case as some small datasets (such as forest-fire) have splits of vastly different complexity.

| Algorithm | CC18 | | CTR23 | |
|---|---|---|---|---|
| | AUC | Accuracy | Correlation | $R^2$ |
| TabDPT | **0.929** [0.927-0.930] | 0.873 [0.872-0.875] | **0.833** [0.823-0.842] | **0.729** [0.712-0.745] |
| TabR | 0.925 [0.922-0.928] | **0.874** [0.871-0.877] | 0.828 [0.814-0.842] | 0.714 [0.692-0.737] |
| XGBoost | 0.925 [0.923-0.926] | 0.868 [0.867-0.870] | 0.827 [0.818-0.837] | 0.711 [0.698-0.724] |
| LightGBM | 0.922 [0.920-0.923] | 0.863 [0.862-0.865] | 0.825 [0.816-0.834] | 0.713 [0.697-0.729] |
| CatBoost | 0.924 [0.922-0.925] | 0.865 [0.863-0.866] | 0.822 [0.808-0.836] | 0.703 [0.682-0.723] |
| PFN++ ($k$NN) | 0.927 [0.924-0.931] | 0.870 [0.868-0.873] | 0.811 [0.799-0.822] | 0.699 [0.686-0.713] |
| MLP-PLR | 0.912 [0.905-0.919] | 0.869 [0.865-0.872] | 0.829 [0.821-0.838] | 0.716 [0.699-0.732] |
| TabPFN ($k$NN) | 0.918 [0.915-0.921] | 0.850 [0.847-0.853] | N/A | N/A |
| TabPFN | 0.898 [0.895-0.901] | 0.812 [0.810-0.814] | N/A | N/A |
| MLP | 0.866 [0.864-0.868] | 0.808 [0.806-0.810] | N/A | N/A |
| kNN | 0.843 [0.839-0.847] | 0.821 [0.818-0.825] | 0.639 [0.626-0.652] | 0.462 [0.445-0.480] |

Table 7: Results on CC18 and CTR23. We report four metrics and their 95% confidence intervals. The best algorithm is bolded for each metric. Furthermore, we underline an algorithm's score if its confidence interval contains the bolded score. TabDPT performs strongly across all metrics on both classification and regression, although regression has much higher uncertainty.

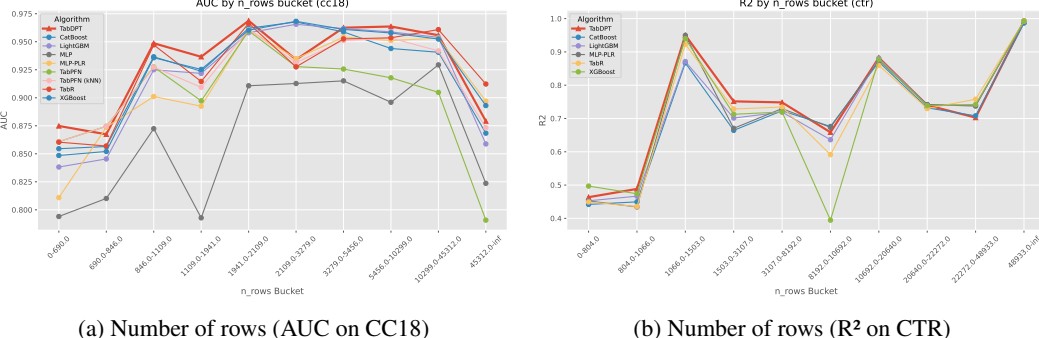

(a) Number of rows (AUC on CC18)  (b) Number of rows (R² on CTR)

Figure 15: Comparison for Number of Rows. For this figure and this figure only, the buckets are quantiles of the data. We can observe that TabDPT's relative performance is slightly lower for larger datasets from CC18.

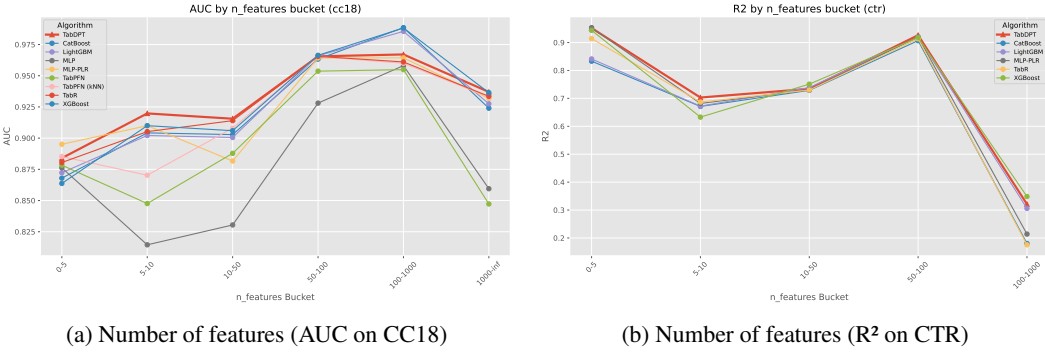

(a) Number of features (AUC on CC18)  (b) Number of features (R² on CTR)

Figure 16: Comparison for Number of Features.

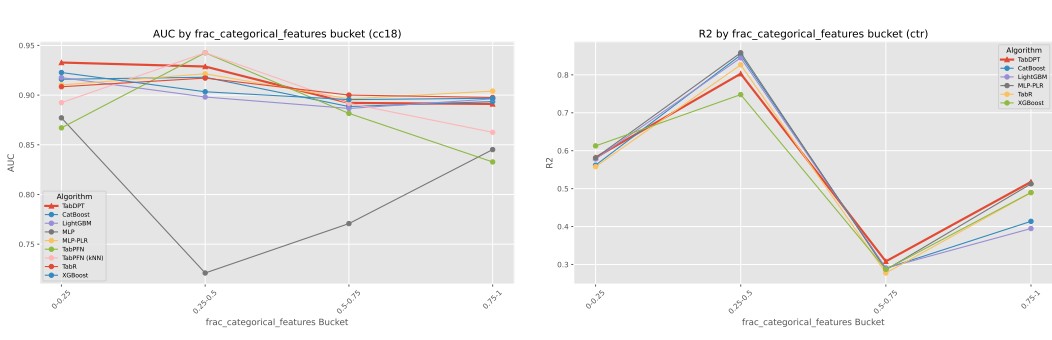

(a) Fraction of categorical features (AUC on CC18)     (b) Fraction of categorical features (R² on CTR)

Figure 17: Comparison for Fraction of Categorical Features.

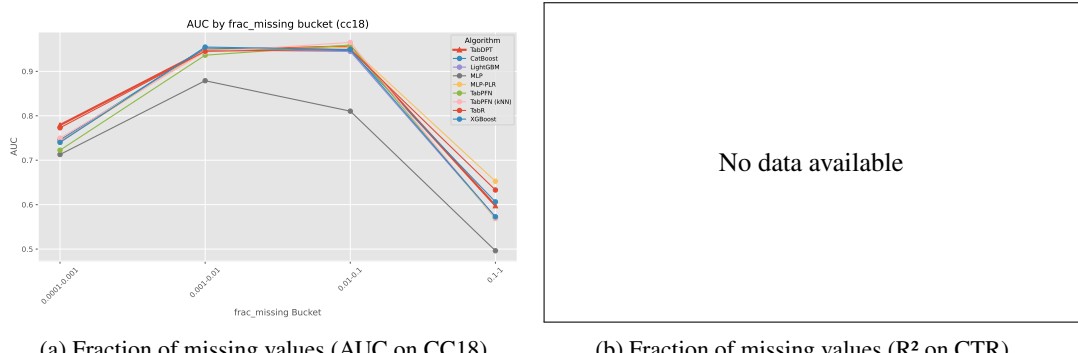

(a) Fraction of missing values (AUC on CC18)     (b) Fraction of missing values (R² on CTR)

Figure 18: Comparison for Fraction of Missing Values. Figure is missing for CTR23 as there are too few datasets with missing values to construct statistically meaningful bins.

## I.3 FEW-SHOT LEARNING RESULTS

We furthermore assess the performance of TabDPT on a unsupervised few shot learning setting. We consider the protocol from STUNT (Nam et al., 2023) on their 10-shot experiment and use the results for STUNT (Nam et al., 2023), CACTUs (Hsu et al., 2018), VIME+LR and ICT available in their paper. Algorithms are evaluated on seven tasks from CC18 and evaluated on accuracy. 10 labelled examples per class are available, the rest of the training is considered unlabelled. TabDPT only using the 10-shots per class performs similarly to the $k$NN baseline. While it is a non-trivial baseline, it is not competitive with modern few-shot methods such as STUNT. However STUNT also uses up to thousands of unlabelled examples for pretraining on each dataset. Note that TabDPT is furthermore rarely trained on small contexts (uniformly sampled between 10 and 1024 during training), so we use a simple method to make use of the unlabelled data and use larger context. We simply predict the class probabilities for the unlabelled training set using the 10 shots as context. Then we take the top-1000 points where our certainty is highest and use them and their predicted labels and the 10 shots as context. This results in TabDPT (semi), a semi-supervised technique using pseudo-labels. This method outperforms STUNT on 5 / 7 datasets and on the average accuracy (averaged over 50 seeds). Furthermore, it requires only forward passes while STUNT requires pretraining for each task.

| Method | cmc | karhunen | optdigit | diabetes | semeion | pixel | dna | Avg |
|--------|-----|----------|----------|----------|---------|-------|-----|-----|
| TabDPT (semi) | 43.46 | **94.17** | 90.20 | 69.00 | **80.23** | **93.93** | 73.99 | **77.85** |
| STUNT | 42.01 | 86.95 | 89.91 | **72.82** | 74.74 | 89.90 | **80.96** | 76.76 |
| CACTUs | 42.14 | 85.48 | 87.92 | 70.75 | 68.22 | 87.21 | 84.40 | 75.16 |
| VIME + LR | 37.92 | 86.63 | 89.63 | 66.56 | 77.66 | 88.71 | 74.73 | 74.55 |
| TabDPT | **43.80** | 90.16 | 88.40 | 68.88 | 74.02 | 88.04 | 65.61 | 74.13 |
| kNN | 41.07 | 85.63 | 87.44 | 71.32 | 74.64 | 87.52 | 71.15 | 74.11 |
| ICT | 38.00 | 88.25 | **90.84** | 67.63 | 74.67 | 89.13 | 69.55 | 74.01 |

Table 8: Accuracy for a 10-shot classification methods across 7 CC18 datasets. The remainder of the training set is accessible but considered unlabelled. While unsupervised meta-learning methods STUNT and CACTUs perform well, TabDPT (semi) achieves higher accuracy on this suite.

## I.4 LARGE DATASETS AND FINE-TUNING RESULTS

On very large datasets, TabDPT's performance can decrease. We hypothesize this is due to the limited context length and the retrieval procedure being less effective on very large sample sizes to build a good "local summary" of the data TabDPT can use. We show nevertheless that finetuning the model can alleviate some of this performance loss on several large datasets taken from Gorishniy et al. (2021).

| Model | CA ↓ | AD ↑ | AL ↑ | EP ↑ | YE ↓ | YA ↓ | MI ↓ |
|-------|------|------|------|------|------|------|------|
| TabNet | 0.510 | 0.850 | 0.954 | 0.890 | 8.909 | 0.823 | 0.751 |
| SNN | 0.493 | 0.854 | 0.954 | 0.897 | 8.895 | 0.761 | 0.751 |
| AutoInt | 0.474 | 0.859 | 0.945 | 0.895 | 8.882 | 0.768 | 0.750 |
| GrowNet | 0.487 | 0.857 | NaN | 0.897 | 8.827 | 0.765 | 0.751 |
| MLP | 0.499 | 0.852 | 0.954 | 0.898 | 8.853 | 0.757 | 0.747 |
| DCN2 | 0.484 | 0.853 | 0.955 | 0.898 | 8.890 | 0.757 | 0.749 |
| NODE | 0.464 | 0.858 | 0.918 | 0.896 | 8.784 | **0.753** | **0.745** |
| ResNet | 0.486 | 0.854 | **0.963** | 0.897 | 8.846 | 0.757 | 0.748 |
| FT-T | 0.459 | 0.859 | 0.960 | **0.898** | 8.855 | 0.756 | 0.746 |
| TabDPT | 0.451 | 0.858 | 0.940 | 0.826 | 8.908 | 0.771 | 0.757 |
| TabDPT (fine-tune) | **0.418** | **0.862** | 0.949 | 0.826 | **8.73** | 0.766 | 0.759 |

Table 9: Accuracy and RMSE for several large datasets from Gorishniy et al. (2021) for different neural network-based baselines. All results except for TabDPT and its fune-tuned version are taken from Gorishniy et al. (2021).

