# OpenReview forum: "TabDPT: Scaling Tabular Foundation Models"
_ICLR.cc/2025/Conference — Submitted to ICLR 2025_

### Official Review · Reviewer_h25x · 2024-10-28

**Soundness:** 3
**Presentation:** 3
**Contribution:** 4
**Rating:** 8
**Confidence:** 5

**Summary:**

This paper proposes TabDPT, a scaled-up version of TabPFN. To achieve this, the authors collect a large number of datasets and then train it by generating a large number of input-output pairs with random columns as outputs. TabDPT outperforms the traditional GBDT on the CC18 and CTR23 benchmarks.

**Strengths:**

1. TabDPT has state-of-the-art performance on both classification and regression tasks. While the previous TabPFN was not applicable to regression tasks, TabDPT proved that this kind of ICL transformer is also suitable for tabular regression tasks, which I think should be appreciated.

2. Using random columns as a useful target feature is a great way to enrich the training dataset.

3. TabDPT's inference time is very efficient because it requires no additional training time.

4. The authors evaluated TabDPT's performance on a variety of datasets to make their results more reliable.

5. It seems reasonable to me to use search for better performance than TabPFN.

**Weaknesses:**

1. While the performance of TabDPT is impressive, the novelty of TabDPT is quite curious. I still think it's a great contribution to the Tabular Learning community, but I think it would be better to emphasize the novelty along with the scale-up part.

2. Some citations are missing. Using random columns as a useful objective feature is similar to masked value prediction in the image or language domains (as the authors say), but it is also a widely used concept in the tabular domain. For example, STUNT [1] and P2T [2] also use this concept to achieve the desired performance on the considered tasks.

----
[1] Nam et al., Few-shot Tabular Learning with Self-generated Tasks from Unlabeled Tables, ICLR 2023

[2] Nam et al., Tabular Transfer Learning via Prompting LLMs, COLM 2024

**Questions:**

1. I would like to know how TabDPT compares to XTFormer [1], even if the training sets are of different sizes.

2. From what I understand, ICL transformers like TabPFN perform better when the dataset size is small (i.e. a few-shot setup). Can you provide a rejection study related to the size of the dataset? Also, it would be great to see a comparison with modern few-shot learning methods like STUNT [2] or FeatLLM [3].

3. I'm also curious about the fine-tuning performance of the model. It is already known that fine-tuning TabPFN can give better results. I wonder if the same phenomenon is true for TabDPT.

----
[1] Chen et al., Cross-Table Pretraining towards a Universal Function Space for Heterogeneous Tabular Data, ArXiv 2024

[2] Nam et al., Few-shot Tabular Learning with Self-generated Tasks from Unlabeled Tables, ICLR 2023

[3] Han et al., Large Language Models Can Automatically Engineer Features for Few-Shot Tabular Learning, ICML 2024

----
Overall, I think this paper will make a high contribution to the ICLR community, and while I still give it an acceptance grade, I am prepared to raise it again if the authors address the concerns noted in the weaknesses and questions.

---

> ### Author Response · Authors · 2024-11-22
> **Answer (1/2)**
>
> We value the time you have taken to review our paper, and your overall positive evaluation. In particular, we appreciate that you noted the high performance of TabDPT, its efficiency during inference, and the extensiveness of the evaluation. We will now respond to your weaknesses and questions below.
>
> ### W1 - Novelty
>
> We are pleased that you have recognized that TabDPT is a quality contribution to the tabular community. When writing the paper, we attempted to be careful not to sell individual aspects as novel – such as the self-supervised ideas or the retrieval in pre-training – but it may be the case that we did not sufficiently sell the novelty of the entire system as a whole. Summarizing that now, all of the individual components which relied on various other sources have been uniquely brought together to form TabDPT, which is the **first technique in tabular data modelling to pre-train on real data and transfer quality predictions to downstream tasks with no additional fine-tuning**. We believe that this opens up a world of possibilities for tabular data!
>
> We agree with you about the “scale-up part” if you have in mind the scaling laws analysis; we have tried to put them forward as a main contribution (Title, figure 1, one page of the paper overall) but it seems you are right as this contribution has generated no discussion in the reviews we received.
>
> We hope that this has answered your criticism here, but we would appreciate it if you could clarify if it hasn’t.
>
> ### W2 - Missing Citations
>
> We are happy to cite STUNT and P2T in an updated version of the manuscript, as they are both relevant and interesting techniques. We will point out however that neither of these methods were able to demonstrate transfer to completely unseen downstream datasets, and so the tasks that they are considering do not exactly map to the foundational setting of our method where we do not assume access to unlabelled data in the same domain for pre-training.

---

> ### Author Response · Authors · 2024-11-22
> **Answer (2/2)**
>
> ### Questions
>
> **Question 1**: While we would also like to know how TabDPT compares to XTFormer, it is difficult to compare with them because (i) their metrics are all relative as opposed to absolute, making it challenging to get a clear picture without re-running their full experimental suite, and (ii) they do not have code available to compare against. Note that we report all of our metrics on an absolute basis whenever possible to at least avoid the first problem, and we plan to release all of our training and evaluation code upon publication to avoid the second problem.
>
> **Question 2**: We have checked the performance of TabDPT as a function of dataset size, and indeed it is the case that TabDPT generally does well when the dataset size is small, and loses a small amount of performance for larger datasets when compared against the baselines.
>
> Here we compare against STUNT. The authors use 7 datasets from CC18 (Census-Income was actually not a CC18 dataset and we trained on it, so we removed it) so that we were able to perform a fair comparison with TabDPT.
> In the table below, which is a copy of Table 2 from the STUNT paper, we added two models TabDPT and TabDPT (semi). In this setting the models are evaluated on 10 shots.
> TabDPT simply uses the 10 shots (and the prototype vector, the average of the given 10 shots, as kNN does; we checked with the authors) as context. While it performs better than kNN, it is not competitive with modern few-shot methods such as STUNT. However STUNT also uses up to thousands of unlabelled examples for pretraining on each dataset.
> Note that TabDPT is furthermore rarely trained on small contexts (uniformly sampled between 10 and 1024 during training), so we use a simple method to make use of the unlabelled data and use larger context. We simply predict the class probabilities for the unlabelled training set using the $k$ shots as context. Then we take the top-1000 points where our certainty is highest and use them and their predicted labels + the $k$ shots as context.
> This results in a TabDPT (semi), a semi-supervised technique using pseudo-labels, which is a very simple way we found to make TabDPT able to use unlabelled data following your comment.
> This method outperforms STUNT on 6 / 7 datasets and on the average accuracy (averaged over 50 seeds). Furthermore, requiring only forward passes we believe it to be much faster than STUNT which requires pretraining for each task.
>
>
> | Method        |     cmc |   karhunen |   optdigit |   diabetes |   semeion |   pixel |     dna |     Avg |
> |:--------------|--------:|-----------:|-----------:|-----------:|----------:|--------:|--------:|--------:|
> | TabDPT (semi) | 43.4576 |     **94.165** |   90.1975 |    69      |   **80.2257** |  **93.93**  | 73.9937 | **77.8528** |
> | STUNT         | 42.01   |     86.95  |    89.91   |    **72.82**   |   74.74   |  89.9   | 80.96   | 76.7557 |
> | CACTUs        | 42.14   |     85.48  |    87.92   |    70.75   |   68.22   |  87.21  | **84.4**    | 75.16   |
> | VIME + LR     | 37.92   |     86.63  |    89.63   |    66.56   |   77.66   |  88.71  | 74.73   | 74.5486 |
> | TabDPT        | **43.7966** |     90.16  |    88.4004 |    68.8831 |   74.0188 |  88.035 | 65.605  | 74.1284 |
> | kNN  | 41.07   |     85.63  |    87.44   |    71.32   |   74.64   |  87.52  | 71.15   | 74.11   |
> | ICT           | 38      |     88.25  |    **90.84**   |    67.63   |   74.67   |  89.13  | 69.55   | 74.01   |
>
> We have not tested against LLM engineered methods as of now, but we would like to stress that LLMs have most probably been trained on many of these datasets and Kaggle notebooks dedicated to each of them. Thus, FeatLLM’s rules could be influenced by memorizing the datasets or examples of feature engineering available online beyond the few-shot examples. As such, these models are very hard to evaluate against fairly.
>
>
>
> **Question 3**: We considered fine-tuning to be orthogonal to the direction of our paper, but we also believe that fine-tuning would indeed improve the performance of TabDPT!

---

> > ### Comment · Reviewer_h25x · 2024-11-22
> > **Thank you for your response**
> >
> > Thanks for your response. I actually really like this work because I agree that training large tabular models that scale to large amounts of data is really challenging, and the authors have successfully scaled it. I think TabDPT can now be seen as a GPT-level foundational model for tabular data.
> >
> > It is also very interesting that the authors extended TabDPT to a semi-supervised learning environment and showed that it outperforms STUNT, which should open the door for more investigation in TabDPT in the tabular learning community.
> >
> > I encourage the authors to reflect on this discussion, and I have raised their score to 8. As the authors note, each of the individual components may not be new, but extending the tabular model while evaluating it on a variety of benchmarks is a great contribution and should be considered very valuable.

---

> > > ### Author Response · Authors · 2024-11-26
> > > **Thank you!**
> > >
> > > We appreciate your positive feedback and the fact that you raised your score. We are happy that you have found value in our model, and we in turn have found this discussion with you and your suggestions to be valuable.
> > >
> > > We hope to continue working on this topic and provide improvements to the model over time as we believe it can have a positive impact within the tabular ML community.

---

### Official Review · Reviewer_jFpp · 2024-11-01

**Soundness:** 2
**Presentation:** 3
**Contribution:** 3
**Rating:** 5
**Confidence:** 4

**Summary:**

This paper aims at presenting an approach which trains tabular-specific In Context Learning -based architectures on real data with self-supervised learning and retrieval. The model is: Tabular Discriminative Pre-trained Transformer (TabDPT). The work is on

**Strengths:**

Efficiency: The model provides fast inference on new tasks due to its ICL capabilities, eliminating the need for task-specific training.

Scalability: TabDPT demonstrates strong scaling with both model size and data quantity, suggesting potential for future improvements.

Generalization: The model generalizes well across tasks without additional fine-tuning, a significant advantage over traditional tree-based models.

Comprehensive Evaluation: The model is thoroughly evaluated against competitive baselines, showing strong performance across various metrics.

**Weaknesses:**

Feature and Class Limitations: The model has predefined limits on the number of features and classes, requiring additional techniques to handle larger datasets.

Textual Information: The current model cannot utilize textual information, which could limit its applicability in certain domains.

Pre-training Cost: While inference is fast, the pre-training process is time and resource-intensive.

Evaluation: I would have expected a larger scale evaluation on tabular data

**Questions:**

1/ How does the model handle datasets with significant missing data, and how does this impact performance?
2/ What are the specific challenges in extending the model to handle textual information, and how might these be addressed?
3/ How does the model's performance compare on datasets with varying levels of feature heterogeneity?
4/ What are the potential applications or domains where TabDPT's approach might be particularly beneficial or limited?
5/ How does the model's performance scale with even larger datasets beyond those tested in the paper?

---

> ### Author Response · Authors · 2024-11-20
> **Thank you for your review: answer (1/2)**
>
> We appreciate your review and that you have recognized our method “efficient” on inference time, “scalable” and “generalizable” to new and unseen data, and that our evaluations are “comprehensive”. We will now respond to your questions and weaknesses one-by-one:
>
> ### Weakness 1: Feature / Class Limitations
> We want to highlight that section 3.4 is specifically dedicated to addressing these limitations. While TabDPT (and many ICL variants of tabular models like TabPFN) are built to handle a limited number of features and predict a fixed set of classes, we bypass this constraint by breaking apart the original prediction task into a new one that TabDPT can handle.
> Let us clearly describe this view for our two methods:
> **Feature length**: To alleviate this problem, we apply PCA to extract the most salient features from the table, reducing the number of features to a manageable size. This transformed dataset is then input into the model, allowing TabDPT to process it effectively and make predictions. On the new figures in appendix you will be able to see shortly that TabDPT performs very strongly, even when there are > 100 features.
> **Class limitations**: We take the original task of predicting a class label that can be large, into smaller tasks (predicting each “digit” of the class label). This effectively allows us to sidestep any restrictions on the number of classes that our model can handle.
> Indeed, we see both of these ideas as crucial contributions to our work, because they allow our method to be truly foundational: being able to deal with tables of various target sizes and column lengths.
>
> ### Weakness 2: Textual information
>
> In fact, we experimented with this idea and found making improvements in this regard is nontrivial. For instance, we observed that the datasets used for pretraining either lacked meaningful textual information or caused the model to overfit to the textual data, rather than learn meaningful statistical relationships in the table. While this remains a promising avenue for future research, we believe improvements along this direction will only be possible with (i) higher quality and (ii) extremely large-scale data to effectively obviate overfitting. We therefore maintain that this is not a straightforward task to accomplish.
>
> ### Weakness 3: Pre-training Cost
>
> While we agree with your perspective, we believe this highlights the core purpose of foundation models: instead of training a separate model for each task, we invest substantial compute resources upfront to pre-train a model. This pre-trained model can then be readily used for predictions without additional training. We would like to highlight our inference time results in Figure 4-a that demonstrates our method is significantly faster when compared to others which require additional training and hyperparameter tuning when faced with a novel downstream prediction task.
>
> ### Weakness 4: Evaluation benchmarks
>
> We respectfully disagree that our evaluations are insufficient, especially since you highlighted them as one of our strengths. Note that CC18 contains 72 datasets and CTR23 contains 35, which totals 107 datasets used for evaluation. This is quite large in the tabular data literature. Furthermore, we only tested on already premade suites so that it was clear no cherry picking was made in the choice of the evaluation datasets.
> That said, if you have a specific benchmark or dataset in mind, please let us know, and we will do our best to incorporate it.

---

> ### Author Response · Authors · 2024-11-20
> **Answer (2/2)**
>
> ### Responses to Specific Questions
>
> **Question 1:**
> Thank you for bringing this up. We have added a figure where we split the datasets based on the fraction of missing data and evaluated our model's performance. We will include this figure in the appendix.  It is possible that our model’s performance decreases more than some of the baseline as the fraction of missing data increases, but the performance remains stable overall.
>
> **Question 2:**
> Please refer to our response to “weakness 2”. In short, (1) high quality data with relevant column labels, and (2) much more extensive tabular datasets are required to achieve this feat as the models tend to overfit to these text labels rather than relevant information.
>
> **Question 3:**
> We kindly ask you to clarify what is precisely meant by “heterogeneity” in the context that you are referring to, so we can better respond to your question. We have added a figure with performance as a function of the fraction of categorical features if you mean heterogeneity in the data type. TabDPT’s performance remains stable overall and performs very well with highly mixed categorical and numerical features. It may decrease slightly as we approach 100% categorical data on CC18, but not on CTR23.
>
> **Question 4:**
> Apart from general-purpose tabular modelling, we believe ICL algorithms can be useful for many applications. For instance, rapid prototyping benefits from this approach, as do scenarios where data is collected in real-time and evolves quickly, yet instantaneous predictions are not necessary. In such cases, retraining a model from scratch repeatedly would be impractical, but predictions that adapt to newly acquired data are still crucial.
>
>
> **Question 5:**
> Thank you for your question. In a similar vein to question 1, we have included our model performance results on different datasets grouped according to their size. We will update the paper with performance broken down by dataset size, number of features and number of categorical features. Our results show that for larger datasets (40k+ instances) TabDPT’s performance decreases slightly below the top algorithms (TabR, XGBoost).

---

> ### Author Response · Authors · 2024-12-01
> **Update**
>
> We wanted to let you know that we recently updated the paper (Appendix I, at the very end) to include some new experimental results requested during the rebuttal period. These results present performance as a function of the number of instances, number of features, ratio of categorical features, and ratio of missing data across both suites of datasets.
>
> Regarding the number of features, we do not observe a decline in performance for TabDPT compared to baseline methods on CC18 and CTR23.
> For the number of classes, 4 datasets in CC18 have more than 10 classes (11, 26, 26, and 46). Computing the accuracy on these datasets (over 10 splits for each method), TabDPT ranks second after TabR, consistent with the results in the main table. While this does not guarantee similar performance with a very high number of classes, we find this result encouraging. Furthermore, one can always use
> $C$-one-vs-all classifiers when handling multiple classes, as most tree-based methods do, instead of our faster $\log(C)$ method. Therefore, we do not consider the number of classes to be a significant limitation.
>
> We also conducted additional experiments, as requested by other reviewers, and found that TabDPT outperforms CACTUs and STUNT (spotlight at ICLR 2023) on few-shot learning tasks using only forward passes. This underscores the versatility and potential of our model, which we believe represents a solid contribution to the tabular data community.
>
> We are pleased that you found our method efficient, scalable, and capable of strong generalization, and that you consider our evaluation comprehensive. We hope our previous responses and the additional experiments have addressed your concerns effectively.
>
> We believe that foundation models designed for and trained on large-scale tabular data represent a promising research direction. Our paper provides detailed insights into training such models on real-world data and demonstrates the scaling laws associated with these approaches. This underscores the value and potential of this research area.

---

### Official Review · Reviewer_9jvJ · 2024-11-02

**Soundness:** 2
**Presentation:** 3
**Contribution:** 2
**Rating:** 5
**Confidence:** 4

**Summary:**

The article introduces TabDPT, a Tabular Discriminative Pre-trained Transformer, designed for tabular data through in-context learning combined with retrieval-based self-supervised pre-training. TabDPT aims to leverage real tabular data rather than synthetic data. The authors demonstrate TabDPT’s state-of-the-art performance on the OpenML-CC18 and OpenML-CTR23 benchmarks for classification and regression tasks.

**Strengths:**

1. This paper is well-organized and clearly written, making it easy for readers to understand the design and details of TabDPT.
2. TabDPT demonstrates scalability with both model size and data size, showcasing the ICL-based model as a foundation model for large-scale tabular pre-training.
3. TabDPT achieves better performance on benchmark datasets like OpenML-CC18 and OpenML-CTR23 while also offering significantly faster inference.
4. The authors openly discuss challenges and "bitter lessons" learned during model development, providing valuable insights and guidance for future researchers in this domain.

**Weaknesses:**

1. The authors present only the direct inference performance of TabDPT. In practical applications, fine-tuning is a reasonable way to enhance performance. It would have been beneficial to compare TabDPT with existing approaches that improve upon TabPFN, such as Tune Tables [1], TabForestPFN [2], MixturePFN  [3], and LocalPFN [4].
2. The novelty of TabDPT appears limited, as it mainly relies on pre-training with real data and adopts the column-as-target approach, a key technique from prior works like STUNT [5], P2T [6], and the KNN-based retrieval strategy used in LocalPFN [4].
3. TabDPT retains certain limitations of TabPFN, such as fixed maximum class and feature counts and a lack of dedicated processing for categorical and textual features. While these issues could be mitigated with conventional methods such as PCA, the authors should provide dedicated evaluations of TabDPT’s performance on datasets with class counts above 10, feature counts exceeding 100, and purely categorical features.

[1] Benjamin Feuer, Robin Tibor Schirrmeister, Valeriia Cherepanova, Chinmay Hegde, Frank Hutter, Micah Goldblum, Niv Cohen, Colin White: TuneTables: Context Optimization for Scalable Prior-Data Fitted Networks.

[2] Felix den Breejen, Sangmin Bae, Stephen Cha, Se-Young Yun: Why In-Context Learning Transformers are Tabular Data Classifiers.

[3] Derek Xu, Olcay Cirit, Reza Asadi, Yizhou Sun, Wei Wang: Mixture of In-Context Prompters for Tabular PFNs.

[4] Valentin Thomas, Junwei Ma, Rasa Hosseinzadeh, Keyvan Golestan, Guangwei Yu, Maksims Volkovs, Anthony L. Caterini: Retrieval & Fine-Tuning for In-Context Tabular Models.

[5] Jaehyun Nam, Jihoon Tack, Kyungmin Lee, Hankook Lee, Jinwoo Shin: STUNT: Few-shot Tabular Learning with Self-generated Tasks from Unlabeled Tables. ICLR 2023

[6] Jaehyun Nam, Woomin Song, Seong Hyeon Park, Jihoon Tack, Sukmin Yun, Jaehyung Kim, Kyu Hwan Oh, Jinwoo Shin: Tabular Transfer Learning via Prompting LLMs.

**Questions:**

1. See weaknesses
2. I’m curious whether the authors could provide a breakdown of TabDPT's performance across different dataset sizes by categorizing datasets into size bins. This would highlight how TabDPT compares with other models at varying dataset scales.
3. TabPFN (subsample) version can efficiently ensemble by sharing context, so a comparison between this method and TabDPT’s retrieval-based strategy in terms of efficiency and effectiveness would be informative.

---

> ### Author Response · Authors · 2024-11-20
> **Answer**
>
> We appreciate your time and effort, and all your constructive comments on our work.  We are pleased that you appreciated our open discussion of “bitter lessons”, and were interested in how our model obeys scaling laws with model size and data size, outperforming models on standard benchmarks purely through ICL without any fine-tuning or downstream training. We will respond to the points raised in the weaknesses and questions in the section below.
>
> ## Weaknesses
>
> ### W1 - Fine-Tuning
>
> One of the key messages in our work is that our architecture can scale with data and get excellent performance, *only through pre-training*, similar to the emergent zero-shot performance observed in modern LLMs. While we agree that fine-tuning and other techniques could improve our performance, it is orthogonal to our main message, and we leave post-training improvements as future exploration.
>
> ### W2 - Limited Novelty
> We agree that using retrieval is not novel in itself (in fact it was used in TabR and can be traced back to local regression about 50 years ago); similar arguments can be made for our self-supervised approach. In fact, we do not claim these as part of our contribution in the introduction of our paper. However, to the best of our knowledge, our work is the first paper to effectively pre-train a tabular foundation model on a large set of real-world datasets, show scaling capabilities akin to ones observed in modern LLMs, and get comparable if not better performance on classic benchmarks.
>
> Furthermore, we are the only tabular method to generalize in this way while pre-training on real data – besides LLM-based techniques such as Tabula-8B, which are *significantly* less performant – increasing TabDPT’s novelty and significance.
>
> Finally, with a mindset of getting the best performance/scaling, we experimented with many different ideas, some of which can be considered novel as mentioned in our “bitter lessons” section, and we only picked the ones that truly made a practical difference.
>
>  ### Questions
> > (2) Breakdown by dataset size & W3: by feature/categories
>
> Thank you for proposing this experiment. We will update the paper shortly with performance broken down by dataset size, number of features, and fraction of categorical features.
> Concerning the performance with respect to dataset size, we observe that for larger datasets (40k+ instances) TabDPT’s performance decreases slightly compared to thebut significantly below the top algorithms (TabR, XGBoost). We will update the paper to reflect this; while we have strong results on CC18, we cannot necessarily expect the method to be as strong on larger datasets. Note that similarly to LoCalPFN [Thomas & Ma et al. 2024], fine-tuning would certainly be an effective strategy to deal with larger datasets, but we were interested in the regime of pure ICL and consider fine-tuning an orthogonal direction which adds to the cost of inference.
> In the updated paper (end of appendix) you will be able to see that our method still performs as well as the best baselines for datasets with a high number of features.
>
> > (3) TabPFN (subsample)
>
> Note that in the paper, “TabPFN (subsample)” is simply called ‘’TabPFN” as this is the default mode of TabPFN; for large datasets, a random context is selected for inference. We can make it clearer by replacing the name “TabPFN” with “TabPFN (subsample)” in the table if you think that would be clearer.
>  As TabPFN uses a smaller transformer backbone, it is slightly faster than our “TabDPT (subsample)” method, however it performs significantly worse. For instance, our “TabDPT (subsampling)” scores an average AUC of 0.912 and Accuracy of 0.84, while TabPFN with subsampling, for the same context size and comparable speed, scores AUC of about 0.9 and Accuracy of 0.812. Our ‘’TabDPT (subsample)’’ with only 512 context size has a faster runtime and higher performance compared to TabPFN with 1024 context size. Thus, whether we use subsampling or retrieval, TabDPT shows superior performance compared to TabPFN.

---

> > ### Author Response · Authors · 2024-11-29
> > **Update**
> >
> > We wanted to inform you that we recently updated the paper, with most of the additional experiments included in Appendix I at the very end. In this section, you will find analyses of the impact of the number of instances, number of features, ratio of categorical features, and fraction of missing values for both suites (CC18 and CTR23) for the models considered in our paper. TabDPT demonstrates overall relatively stable performance (no significant drops), particularly with the number of features (above 100 and even 1000) for the datasets considered. However, it does exhibit slight signs of decline for larger datasets or a higher ratio of missing data.
> >
> > We also added experiments comparing our method to STUNT in a 10-shot learning setting. TabDPT outperforms STUNT using only forward passes, which are extremely fast compared to most unsupervised meta-learning approaches. Additionally, we tested our method on several very large datasets (up to 1.2 million samples) and showed that fine-tuning helps the model scale effectively to these large datasets.
> >
> > We apologize for the oversight regarding “Weakness 3.” The part about the number of features is addressed above and in Fig. 16. Regarding the number of classes, four datasets in CC18 have more than 10 classes (11, 26, 26, and 46). Computing the accuracy on these datasets (over 10 splits for each method), TabDPT ranks second after TabR, consistent with the results in the main table. While this doesn’t guarantee similar performance with a very large number of classes, we find this result encouraging. Furthermore, it is always possible to employ C one-vs-all classifiers when handling multiple classes, as most tree-based methods do, rather than using our faster log(C) method. Therefore, we do not consider the number of classes to be a significant limitation.
> >
> > We thank you again for your review and for acknowledging the strengths of our paper, including the writing, scalability, achieved scores, potential positive impact and the detailed technical explanations.
> >
> > We hope all your concerns (fine-tuning, novelty, number of classes/features) have been addressed in these answers. We are very excited about the potential of this research direction and believe it offers significant contributions and insights. We would sincerely appreciate your consideration of this additional evidence, and we hope it strengthens your confidence in the significance of our work.

---

### Official Review · Reviewer_HS6o · 2024-11-03

**Soundness:** 3
**Presentation:** 2
**Contribution:** 2
**Rating:** 3
**Confidence:** 5

**Summary:**

This paper provides an in-context learning (ICL) scheme TabDPT for neural networks (NNs) on tabular prediction tasks by pre-training a shared Transformer backbone and making predictions with labeled-neighborhood context in a row-based encoding manner across open-domain classification or regression tabular datasets. Specifically, during pre-training approximate retrieval strategy is used to fetch neighbors as the contexts for given data points, self-supervised learning is performed by further dividing them into context and query splits and reconstructing the selected target column features, fitted with randomly shuffled order and masked values of other features. During inference, exact retrieval strategy is used for each query data point to form its labeled-neighbor context for direct prediction. Pre-trained on 123 open-domain datasets (32M rows and 2B cells) from OpenML, the evaluations on two public benchmarks (CC18 for classification and CTR23 for regression) show TabDPT can be comparable to recent supervised tabular NNs and traditional GBDTs, achieved without training on downstream datasets. The scaling behavior of TabDPT in both model parameters and pre-training data size is explored.

**Strengths:**

**Novel scheme & data scenario combination**: Existing in-context learning (ICL) schemes in tabular prediction community are mostly based on LLM backbones and focus on few-shot or zero-shot data scenarios, while TabDPT is a non-LLM tabular ICL scheme on fully labeled downstream datasets. Principally, TabDPT is pre-trained to learn to predict by comparing with labeled neighbors rather than LLM-based schemes that may rely on world knowledge in the LLMs.

**Robust performance comparison, analysis & ablation**: The authors offer result confidence intervals, win-rate matrix and Elo score analysis on evaluated benchmarks, which give a clear and robust performance comparison between TabDPT and other baselines. The ablation study shows the sources of main bonus in TabDPT scheme.

**Detailed limitation discussion**: The authors sufficiently discuss the limitations of TabDPT caused by its inherent design and the special nature of tabular data features.

**Weaknesses:**

**Limited technical novelty & consideration in the field**: To my knowledge, the most technical components in TabDPT scheme is not novel in common tabular data learning community.
- In Sec. 3.1 the authors propose: (1) Row-based encoding strategy (Line 147,155) to reduce memory consumption for in-context training, and do not treat categorical or numerical variables differently (Line 144), while row-based encoding is a common practice in multiple-table prediction or graph-based tabular models (e.g., RDL [1]), and finely distinguishing numerical, categorical, binary and other tabular cells is beneficial [2]. (2) Shared Transformer backbone is a common design choice in cross-table learning that TabDPT is similar to the pre-trained tabular neural networks like TranTab [2] and XTab [3].

- In Sec. 3.2 the authors propose a self-supervised approach to pre-train TabDPT with (1) Random Column as Target prediction and (2) Column Shuffling and Masking inspired by the NLP masked language modeling, **while constructing the random column and masking input columns are old and very common in traditional pre-training objectives for tabular deep learning [4]**. Besides, **both technical components (including column shuffle) are widely used in LLM-based tabular model pre-training** (e.g., TapTap [5], GReaT [6], CM2 [7]).

- In Sec. 3.3 the authors propose retrieval-based strategy for pre-training TabDPT, sharing a similar training strategy as retrieval-based tabular models (e.g., RETRO or TabR [8] as mentioned in Line 211), though using ICL prediction manner, the core difference of TabDPT is solely substituting kNN search with more memory-efficient retrieval algorithm used in [9], which is not novel as well.

In summary, from technical novelty perspective, TabDPT seems to be a combination of existing works in tabular learning community, with the similar overall framework in other LLM-based tabular ICL papers, which weaken the original contributions of the paper.

**Insignificant performance promotion**: TabDPT performed heavy pre-training on 123 datasets (2B cells) and also requires fully labeled training data to form neighborhoods' contexts to conduct ICL-based prediction for a given data point, its main performances in Table 1 seem not significantly different from supervisedly tuned baselines like recent retrieval-based deep learning method (i.e., TabR) and classical tree-based models, the performances are so close that may be changed by selecting proper random seeds, raising a question of whether it is necessary to adopt ICL-based scheme under such fully labeled tabular data scenarios.

**Unreasonable computational budget comparison**: In Sec. 5.3 the authors discuss the training and inference time of TabDPT and other supervisedly tuned baselines (see Fig. 4a). There seem to be two perspectives to reflect the partially unreasonable analysis here: (1) In real-world practice, for a tuned supervised baseline, inference time is the most important efficiency metric since a model is only tuned once but used to predict in the long term, thus a direct comparison just using inference time of TabDPT and other baselines is more convincing and practical. (2) If the authors want to compare the development time of TabDPT and others, the pre-training time may need to be recorded and considered since this part is also the training budget of TabDPT, comparing only inference time of TabDPT with training (HPO) + inference time of others may hinder the real computational requirement comparison. **In summary, from any perspectives above, the computational budget analysis may be not reasonable enough**. Besides, the authors compared convergence speed of TabDPT and PFN++ using a fixed training epoch, while the trend may be affected by hyperparameter settings, and comparing under a fixed training time may be more rigorous.

**Limitations from pre-fixed maximum feature amount and class number**: As discussed in Sec. 3.4 and Sec. 6, TabDPT has pre-fixed maximum feature amount and class number to process, which inherently limit its efficiency and effectiveness in long tables (commonly seen in recommendation field) or large class number. For long tables, dimensionality reduction techniques should be applied to inevitably protect the input features. For large class number, multiple forward time is required which further add inference budget and may be hard to fit.

**Uneconomical inference strategy**: Compared to the traditional supervisedly tuned baselines (i.e., tree models, deep learning models), the retrieval-based nature of TabDPT may hurt its real practicality in industrial tabular data scenario where the labeled data scale is extremely large (in both sample and feature amounts) and online real-time application is required.


**Reference**

[1] Position: Relational Deep Learning - Graph Representation Learning on Relational Databases, ICML 2024.

[2] Learning Transferable Tabular Transformers Across Tables, NeurIPS 2023.

[3] XTab: Cross-table Pretraining for Tabular Transformers, ICML 2023.

[4] Revisiting Pretraining Objectives for Tabular Deep Learning, Arxiv 2022.

[5] Generative Table Pre-training Empowers Models for Tabular Prediction, EMNLP 2023.

[6] Language models are realistic tabular data generators, ICLR 2023.

[7] Towards Cross-Table Masked Pretraining for Web Data Mining, WWW 2024.

[8] TabR: Tabular Deep Learning Meets Nearest Neighbors, ICLR 2024.

[9] Retrieval & fine-tuning for in-context tabular models, Arxiv 2024.

**Questions:**

Honestly, it is interesting to see ICL-based inference scheme can be comparable to traditional supervised scheme with prompt-tuning-like pre-training and sufficient neighbor contexts. I would like to improve my score according to the response of the following questions and comments from other reviewers.

(1) Since a 78M TabDPT pre-trained on 2B cells is used for the main results, and the author claimed there is no single gold standard benchmark in the paper, could you further evaluate on the following datasets: (a) the ones in the paper of FT-Transformer [1] (7 classification & 4 regression datasets), (b) several datasets from "Categorical classification" & "Categorical regression" in Appendix A.1 of [2]. I would be more familiar with these deep-model- or tree-model-favored datasets (including large feature and class amounts). The results of TabDPT are enough, and add the total or per-sample inference time if possible.

(2) Since different tabular datasets may vary in feature ranges, is there any consideration or experiment to reflect TabDPT is able to handle datasets in various feature ranges? (e.g., a table with feature value range from 1\~10 and another from 1,000\~100,000 in a single batch)

(3) Does TabDPT design considered the semantics of column names? What about its performances in OOD (out-of-domain) downstream datasets (i.e., the results on the datasets which domain is not pre-trained)?

(4) According to Fig. 4b, the performance of TabDPT is heavily rely on the neighbor retrieval during inference, forming a similar mechanism of kNN. Is it possible to substitute TabDPT with the kNN having a learnable neural kernel?

(5) Could you provide a comparison of inference time per 1000 samples for TabDPT and other baselines (i.e., only record inference time for others in Fig. 4a)?

(6) Since ICL-based outputs are affected by contexts, i.e., retrieved neighbors in TabDPT, is there any consideration to keep the stable prediction (especially regression tasks), or will the results be hugely changed with different random seeds?


**Reference**

[1] Revisiting Deep Learning Models for Tabular Data, NeurIPS 2021.

[2] Why do tree-based models still outperform deep learning on tabular data? NeurIPS 2022.

---

> ### Author Response · Authors · 2024-11-19
> **Answer (1/4)**
>
> We would like to thank you for the time you took to review our paper and asking pertinent and precise questions. We will include your feedback to improve our paper and will address the points you raised below.
>
> We will group your concerns into four main categories 1) General points about ICL models for tabular data, 2) Our contribution, and 3) Evaluation and limitations and 4) additional unaddressed technical questions.
>
> ## General points about ICL
> > Weakness about unreasonable computational budget and Uneconomical inference strategy
>
> ### Pretraining time:
> The point about including the pre-training time of ours is a fair question to ask. We think these are different perspectives which are common to all foundation model-type works, not just ours.
> As we released our model, our intention is for people to directly use the pretrained model when faced with a new task. So if you would like to use our model on a new dataset, and our models take 10min to produce results, is the training time 10min or 10min + about a week of computation?
> There are valid reasons to consider the latter, for instance if we are concerned with CO2 emissions. From a user perspective, only the inference cost is paid, furthermore if the user (or ensemble of users) test on a great number of tasks, the total computational cost could be considered to be amortized across tasks. For instance, let’s call the fixed pretraining cost $T$, the average task-specific time for TabDPT $t$, and the number of tasks $n$. If the XGBoost total train+test on a task is on average $c \times t$, with $c>1$, then for enough tasks (i.e., large enough $n$), we will have $c \times n \times t > n \times t + T$).
> Let us know if we understood and addressed your point correctly.
>
> ### Inference time
> Yes, you raise valid points and we will update the paper to reflect them. First, we absolutely agree that inference time is key in many industrial applications. We think there is and will always be a place for algorithms with a fast inference time.
> That being said, we also think there is a place for algorithms that have a much lower training+inference time overall even if the inference time alone is slower. A simple example is rapid prototyping. More interesting examples are cases where the data is gathered in an online manner and changes quite fast, but our predictions do not have to be instantaneous – you would not want to train an entirely new model every time, but you would still like your predictions to be more adaptive to the newly-acquired data.
> We can consider for instance marketing/content recommendations applications, where what a user clicked on during the day or what ads/marketing campaign they were exposed to should have a big influence on what is recommended to them in the following hours/days. Using a fixed model could be very problematic here, as it would need to be retrained every time. We do not wish to recommend a product the client just bought, for instance, so the model needs to adapt to every user data every day. This would be challenging for more classical models but we think is very suitable for our type of model.
> We will update our paper to reflect this and provide inference time comparisons with classical models (they are indeed a couple orders of magnitudes faster).
> Lastly, there are numerous methods to improve inference speed of pretrained transformers (sometimes at the cost of performance, sometimes not) such as ONNX, specialized hardware or quantization. We have not looked into those methods though as we consider this somewhat outside the scope of the paper.

---

> ### Author Response · Authors · 2024-11-19
> **Answer (2/4)**
>
> ## Our contribution
> > Limited technical novelty & consideration in the field
>
> We would like to start out by saying that we agree with a fair amount of what you said, but also that you are perhaps missing the most novel points of our paper by focusing on points that we agree are not novel. In particular, we agree that training with retrieval for tabular data is not new, dating back to at least the 1970s with locally weighted regression (Williams 1979). Transformers with rows-as-tokens have also been used in TabPFN and some citations you additionally provided, and indeed masking-based SSL has already existed in many fields, including some works in tabular data (and it can even be traced back to Gibbs sampling!). Our contribution has never been bringing these tools to tabular data, but we are happy to add citations to the works you have noted and can attempt to be even more clear in the writing that we do not consider these individual points to be our main novelty.
>
> Given the examples provided in the Inference time section, we argue that large in-context tabular models have useful applications. We are quite excited by this line of research, but outside of LLM-based models that perform poorly, and alternative cross-table training techniques such as XTab that require downstream fine-tuning, only TabPFN (and some of its variants) provide quality predictions out-of-the-box with no further weight updates. However, the latter is only trained on synthetic data, and the few other papers retraining a similar model also did it on synthetic data [ForestPFN].
>
> We are interested in being able to train such models on real data for two reasons: 1) it might not be obvious how to scale the synthetic prior, and we know from the foundation models literature that scaling the data is key for performance; and 2) we are interested in being able to train on large amount of real data that may or may not be public. For example, consider a large company having lots of internal data: being able to train such a model on its own in-distribution data is simpler than editing that the TabPFN prior in a way that it captures this distribution.
>
> Considering these scenarios, our goal becomes to train a tabular model using real data in a way that transfers to downstream tasks without fine-tuning (although we acknowledge that fine-tuning is indeed likely to improve performance; we consider that orthogonal to the direction of the paper).
>
> Here are several realizations we had throughout the course of creating TabDPT: 1) many datasets are needed, but not as many as we had thought initially, 2) only using a supervised target leads to fast overfitting, even with a large number of tables, 3) mixing datasets in the batch is important, 4) doing random target selection and column dropping is key for efficient utilization of the data (it can be seen as an analog to next token prediction), 4) retrieval does help, 5) a lot of optimization choices end up mattering, 6) a lot of encoding/architectural choices (surprisingly) end up NOT mattering, 7) we can perform classification and regression with a single model, but for best performance the two tasks should be shared as much as possible in the model to make better use of the data, 8) the quality of the data matters, which is why we preferred sourcing from openML rather than CommonCrawl (many small tables.)
> One of our contributions is sharing the lessons we learned on how to use real data from many different sources to train a tabular foundation model.
>
> Our second main contribution is providing scaling laws for both model size and data amount. We would appreciate any discussions regarding this contribution.. To the best of our knowledge, this is a first in the tabular domain. The fact  that tabular foundation models would exhibit scaling properties similar to LLMs, is not well-established in tabular data literature. We also provide joint data and model size scaling laws, and show how to use real data to unlock it.
>
> These points above constitute what we see as our main technical contributions. Furthermore, we provide evaluations, another way to compare methods with Elo scores (standard in LLM arenas but far from the standard in tabular data). The motivation for using these types of metrics in the LLM domain is to sort them based on human rankings, while in our setting it unlocks ranking models on a group of datasets without having to evaluate all of them on every single dataset. Finally, we release model weights (and eventually full training code).

---

> ### Author Response · Authors · 2024-11-19
> **Answer (3/4)**
>
> ## Evaluation and performance
> > insignificant performance promotion and (1) additional results
>
> First, we would like to clarify that our evaluation contains confidence intervals over different splits of the data. While many papers in the literature fix the split and only rerun the algorithm, which is insufficient to capture real uncertainty.
> Thus, most of the uncertainties come from the predefined splits having different levels of complexity.
> Here we report a new table containing the Interquartile Mean (IQM) scores instead of the mean scores. This is a recommendation from “Deep RL at the edge of the statistical precipice” (NeurIPS best paper award) which recommends using IQM instead of mean to better estimate confidence intervals when the number of seeds is limited, and has become somewhat of a standard for tabular data.
> As you can see, IQM, which is the mean of the score discarding the lowest 25% and highest 25% of scores, shows lower confidence intervals.
>
> | Algorithm | AUC (CC18) | ACC (CC18) | R2 (CTR23) | CORR (CTR23) |
> |-----------------|-----------------------|-----------------------|-----------------------|-----------------------|
> | **TabDPT** | **0.972 ± [0.971, 0.973]** | 0.917 ± [0.915, 0.919] | **0.831 ± [0.826, 0.835]** | **0.911 ± [0.908, 0.913]** |
> | TabR | 0.967 ± [0.965, 0.969] | **0.923 ± [0.920, 0.926]** | **0.825 ± [0.818, 0.831]** | **0.909 ± [0.905, 0.912]** |
> | MLP-PLR | 0.967 ± [0.965, 0.968] | 0.914 ± [0.911, 0.917] | **0.827 ± [0.822, 0.832]** | **0.907 ± [0.904, 0.910]** |
> | PFN++ (kNN) | **0.970 ± [0.968, 0.972]** | 0.913 ± [0.910, 0.916] | 0.792 ± [0.782, 0.801] | 0.888 ± [0.881, 0.894] |
> | XGBoost | 0.966 ± [0.964, 0.967] | 0.911 ± [0.909, 0.913] | 0.820 ± [0.814, 0.825] | 0.904 ± [0.900, 0.907] |
> | LightGBM | 0.962 ± [0.960, 0.964] | 0.908 ± [0.906, 0.910] | 0.809 ± [0.803, 0.815] | 0.900 ± [0.896, 0.904] |
> | CatBoost | 0.959 ± [0.958, 0.961] | 0.903 ± [0.901, 0.905] | 0.802 ± [0.794, 0.810] | 0.897 ± [0.890, 0.903] |
> | TabPFN (kNN) | 0.959 ± [0.955, 0.962] | 0.884 ± [0.881, 0.887] | N/A | N/A |
> | TabPFN | 0.939 ± [0.935, 0.943] | 0.852 ± [0.849, 0.855] | N/A | N/A |
> | MLP | 0.910 ± [0.907, 0.913] | 0.863 ± [0.860, 0.866] | N/A | N/A |
> | kNN | 0.874 ± [0.869, 0.879] | 0.866 ± [0.862, 0.871] | 0.466 ± [0.446, 0.485] | 0.671 ± [0.654, 0.687] |
>
> As it is evident from this table, TabR/MLP-PLR and TabDPT show strong performance above the rest of the algorithms on CTR23. On CC18, TabDPT performs significantly better in terms of AUC, but is outperformed in terms of accuracy by TabR. In all cases the confidence intervals are smaller, differentiating the top algorithms from the rest more reliably.
>
> ### Additional experiments per dataset size and number of features.
> We will update the paper with the figure of the performance of the different algorithms vs. dataset size or number of features.
> To give a summary of the new findings: TabDPT is stable with a high number of features (even > 100 and > 1000), comparable to how TabR and MLP-PLR behave. This underscores that even though there are some constraints on the size of the datasets TabDPT handles during training, it still generalizes well during evaluation. Note that other algorithms (such as TabR, LightGBM, CatBoost) struggle in terms of runtime on datasets with a large number of features.
>
> However, with respect to the number of instances, we observed a drop in performance when dataset size is larger than 40k. In that regard, CC18 and CTR23 – which contain both small and large(r) datasets – are beneficial to TabDPT. We will include this finding in the paper and specify that while TabDPT shows strong performance on CC18 and CTR23, it may not perform as well on very large datasets (without fine-tuning, at least as shown in LoCalPFN [Thomas et al., 2024]).
>
> Additionally, we will provide results for the large datasets you mentioned, along with the categorical classification and categorical regression benchmarks, but first we need to filter them as some have been used for pretraining. We can probably expect TabDPT’s performance to be lower than the top performing algorithms on the very large datasets.

---

> ### Author Response · Authors · 2024-11-19
> **Answer (4/4)**
>
> ## Additional questions
> > number of features
>
> Note that our choices are based on the test datasets we have, i.e. increasing the number of features from 100 to 1000 might let us encode 5 more datasets without subsampling, however as more training and test data becomes available we plan to increase these numbers to cover more datasets. We also tried 256 and 512 maximum features without much of an impact on the overall performance. Furthermore, while this architecture indeed does have an exact maximum number of features, even algorithms that can theoretically handle an arbitrary number of features eventually struggle. CatBoost, LightGBM, and TabR specifically showed training times that increased dramatically with the number of features, taking more than 5 hours on some datasets with many features.
> > Number of classes
>
> As we end up predicting the digit number, we can, in theory, produce predictions for any number of classes C and the inference cost increases by a factor of O(log C). Note that many algorithms (including tree based ones) use a one-vs-all type of predictions leading to a scaling O(C). Only 3 datasets on CC18 have more than 25 classes (up to 46), we do not observe a loss of performance for TabDPT but further investigations on other datasets would be needed to confidently answer.
> > (2) feature ranges
>
> All tables are processed independently so different feature ranges would not affect other tables. It is possible that a table with a large feature range would be harder to predict. However we did try signed $log(1+x)$ pre-processing of the features as well and did not observe any significant difference on the evaluation.
>
> > (3) column names
>
> We did actually try this: we encoded in a simple way information about column names in the embedding based on fasttext as a first test. However we observed that while our training loss was significantly lower, our test loss was higher. In short, while we have many “cells”, we don’t have that many features (~60 features on average over ~120 tables). In this case, using column names leads to the model overfitting. One of our main motivations for using real data is actually using column names but we believe we need a lot more data and potentially additional augmentations (like masking feature names, etc..)
>
> > (3) out-of domains datasets
>
> The general domain names we used are very broad and as such most datasets can fit into one of the categories. We can provide ranks here for TabDPT, XGBoost and TabR for a sample of datasets that appear very different from the rest.
> On Tic-Tac-Toe a game dataset, we have TabDPT (rank 3 on AUC, 2 on ACC), TabR (rank 1 on AUC, 3 on ACC), and XGBoost (rank 2 on AUC, 1 on ACC). And onFirst Order Theorem Proving (mathematics), we have TabDPT (rank 3 on AUC, 1 on ACC), TabR (rank 5 on AUC, 5 on ACC), and XGBoost (rank 4 on AUC, 3 on ACC). We can additionally provide the csv containing all evaluation results for all datasets on all folds.
>
> > (4) TabDPT depends on kNN: can we learn the kernel?
>
> This is a very good question. We have tried several things in the past such as using the kNN on the embeddings or first key/value embeddings. This did not lead to any performance improvement. Note that using any deeper embedding is problematic for ICL models as the embeddings themselves depend on the context, so there is the question of which context to use for embeddings.
> To learn a kernel for kNN there are two main principled ways we thought about: (1) is to learn a kernel through zero-th order optimization. In our experiments, while it can work on simple datasets it is very hard to optimize. (2) There are methods for relaxing kNN into a continuous problem, however these methods would add a significant computational burden.
> Furthermore, it is complex to adapt (1) and (2) efficiently in the ICL/multi-table setting where the kernel would have to be dataset-dependent.
>
> > (5) Inference time
>
> We can add precise numbers and we will be happy to include that information in the text or in the figure depending on your feedback. CatBoost/XGBoost have about 0.01-0.05s/1000 samples inference time in some earlier tests we ran using Tabzilla’s data. This is significantly faster than TabDPT, but as mentioned earlier, whether this number or the “time to prediction on a new table” matters depends on the specificities of the problem.
>
> > “(6) Since ICL-based outputs are affected by contexts, i.e., retrieved neighbors in TabDPT, is there any consideration to keep the stable prediction (especially regression tasks), or will the results be hugely changed with different random seeds?”
>
> Could you clarify the question? In our work we actually use exact neighbour computation during inference, as the bottleneck is the transformer forward pass rather than the search, and thus the model is deterministic (as it is also permutation invariant). However, for very large datasets we might require an approximate retrieval scheme which can depend on a random seed.

---

> > ### Comment · Reviewer_HS6o · 2024-12-02
> > **Thank you for the concrete response**
> >
> > I sincerely appreciate the concrete and patient response from the authors, which has answered my questions and located the main concerns, i.e., three points, (1) computational budget, (2) technical novelty in the field, and (3) performance significance in the weaknesses. I glad to see the authors express considerable agreement on the mentioned weaknesses and give detailed explanations:
> >
> > **Feedback for Answer 1 on concerning computational budget**: Sounds good, the authors honestly acknowledge the pre-training cost is not taken into account when comparing computational budget with classical supervised baselines since TabDPT is positioned at a foundation model for tabular prediction like ChatGPT for language tasks. Also, the inference time was not compared, TabDPT's retrieval-based inference requiring extra time for neighborhood searching operation, which posing a major limitation for application. Although, as the author replied, such inference process by pre-trained model may be accelerated using ONNX, the baselines can also become more efficient with the same acceleration techniques, the inference time limitation is always here. From perspective of foundation model, ChatGPT is widely recognized due to its remarkable performance lead, even compared to supervised models, rather than it is designed in a foundation manner, while the performance of TabDPT seems insignificant compared to supervised baselines, in which case the computational budget becomes a major concern point, otherwise, why we do not use the traditional methods?
> >
> > **Feedback for Answer 2 on limited technical novelty**: The authors fairly summarize their technical contributions, i.e., (1) training a tabular model using real data in a way that transfers to downstream tasks without fine-tuning, (2) providing scaling laws for both model size and data amount. In the first point, there are previous works like in-context tabular model with synthetic data, retrieval-based supervised model but require fine-tuning (e.g., TabR), pre-training on real data but LLM-based model, **it sounds the seemingly technical contribution is a new combination of previous aspects, and the TabDPT performance is not leading enough to match the position of "foundation model" like ChatGPT in language tasks**, given the current results on the evaluated data. For the second point, even in tabular data fields, **previous works of LLM-based tabular model pre-trained on real data have partially demonstrated the conclusion, which is also non-novel**.
> >
> > **Feedback for Answer 3 on insignificant performance**: The authors list detailed performance analysis in Answer (3/4) and argue from the perspective of Interquartile Mean (IQM) scores instead of the mean scores, the results show TabDPT, though has statistically better performance on evaluated data, is stably close to the top baselines on the evaluated datasets. On CC18, TabDPT performs significantly on AUC, but is outperformed on accuracy by TabR. **Considering the close performance with relatively heavy training and inference computational cost**, my basic impression on the work is not changed.
> >
> > However, excluding the essential concerns, the exploration on the in-context tabular prediction model is really encouraging and beneficial, and the response is clear enough to answer my questions point by point. **My value and concerns are relatively practical and result-oriented, which only represent my personal opinion. I think the meta-reviewers can comprehensively refer to the feedback from other reviewers to make a tradeoff for the final decision.** Although I would like to hold my consideration, I agree that **the research direction is great**.

---

> ### Author Response · Authors · 2024-11-27
> **Additional response**
>
> Following recommendations by other reviewers, we analyzed the score as a function of the dataset size and we realized that there is a drop in TabDPT's performance as dataset size grows (Fig 15 in Appendix).
> This is confirmed on the larger datasets you asked us to consider. Note that in hindsight, this may not totally be a surprise as we rely on kNN to build our context. While this improves significantly over random samples, kNN still might return lower quality neighbourhoods if the dataset is very large, no matter how good the model is.
> In Fig 4a of LoCalPFN, the performance of TabPFN+kNN does not scale as well with dataset size when compared to strong tree-based baselines. But LocalPFN and MixturePFN noticed that performing finetuning can improve results significantly, so we also report results with fine-tuning for TabDPT. Note that we did not perform any hyperparameter optimization and used the same default setting for all datasets, and so we are confident we could further improve on those results.
>
> Here is Table 2 from ‘’Revisiting Deep Learning Models for Tabular Data’’ updated with our results. We removed 4 datasets that were present in our training set (see Table 2 of our paper). We see in the table below (on neural network based models), on the datasets from the FT-Transformer paper, that a fine-tuned TabDPT is able to achieve the best results on 3 out of 7 tasks (using simple, fixed hyperparameters for all datasets). While this is not fundamental to the original aim of the paper in our opinion, it demonstrates that TabDPT can be adapted to more challenging tasks with some additional effort.
>
> | Model              |   CA ↓ |   AD ↑ |    AL ↑ |   EP ↑ |   YE ↓ |   YA ↓ |   MI ↓ |
> |:-------------------|-------:|-------:|--------:|-------:|-------:|-------:|-------:|
> | TabNet             |  0.51  |  0.85  |   0.954 | 0.8896 |  8.909 |  0.823 |  0.751 |
> | SNN                |  0.493 |  0.854 |   0.954 | 0.8975 |  8.895 |  0.761 |  0.751 |
> | AutoInt            |  0.474 |  0.859 |   0.945 | 0.8949 |  8.882 |  0.768 |  0.75  |
> | GrowNet            |  0.487 |  0.857 | nan     | 0.897  |  8.827 |  0.765 |  0.751 |
> | MLP                |  0.499 |  0.852 |   0.954 | 0.8977 |  8.853 |  0.757 |  0.747 |
> | DCN2               |  0.484 |  0.853 |   0.955 | 0.8977 |  8.89  |  0.757 |  0.749 |
> | NODE               |  0.464 |  0.858 |   0.918 | 0.8958 |  8.784 |  **0.753** |  **0.745** |
> | ResNet             |  0.486 |  0.854 |   **0.963** | 0.8969 |  8.846 |  0.757 |  0.748 |
> | FT-T               |  0.459 |  0.859 |   0.96  | **0.8982** |  8.855 |  0.756 |  0.746 |
> | TabDPT             |  0.451 |  0.858 |   0.94  | 0.826  |  8.908 |  0.771 |  0.757 |
> | TabDPT (fine-tune) |  **0.418** | **0.862** |   0.949 | 0.826  |  **8.736** |  0.766 |  0.759 |
>
>
> Here are some of the changes we made that may interest you:
> - (Appendix) Large datasets results with discussion about performance on large datasets and how fine-tuning might help.
> - (Appendix) Few-shot learning: While not requested in your review, another reviewer asked about few-shot learning; we found that we were able to **match strong meta-learning baselines using only forward passes**, so we included this result as well.
> - (Main text) Runtime: We added in blue in 5.3 an explanation of the point you made
> > However, while tree-based and DL baselines offer faster inference after training, their efficiency depends on the scenario: TabDPT is advantageous when frequent retraining is necessary, whereas traditional models are preferable for fixed data needing rapid inference.
>
> While we are limited by space, we are willing to make efforts to include more statements/discussions in the paper on points you raised. For instance we can add the *inference-only* time for the baselines in Fig 4a, thus showing more clearly the tradeoff, and another discussion referencing the finding on larger datasets.

---

> ### Author Response · Authors · 2024-12-03
> **Answer (1/2): Faster inference**
>
> **We genuinely thank you for replying to us and articulating your concerns clearly, which we greatly appreciate. You have a strong technical understanding of our method and its limitations, and we would like to continue the discussion and share more of our thoughts and quantitative results on the main points you raised.**
>
> ### Performance/Speed Trade-off and 4x Inference Speed Improvements
>
> We continue the discussion here on both points 1 and 3. We are pleased that you recognize that TabDPT is a strong method, and we hear your concern about the inference speed.
>
> We agree that in a typical scenario where a training set is given, a model is trained and then deployed to be used for a long period of time, strong tree-based methods are preferable. However, in cases such as (1) non-stationary data where the model needs to be retrained often, (2) few-shot learning (where we outperform STUNT, a spotlight paper at ICLR 2023), and (3) smaller datasets in general, TabDPT could be faster and/or better.
>
> **We argue that the very fact that TabDPT has different strengths and weaknesses compared to classical methods makes it more likely to be used in practical scenarios.**
>
> To be fully transparent, we initially thought that our model would only perform well when fine-tuned on datasets larger than a few thousand samples, similar to TabPFN and the findings of LocalPFN/MixturePFN. We were pleasantly surprised to see we could rival strong methods on entire suites without fine-tuning, with a total per dataset time budget that is orders of magnitude lower than baselines. This explains why we did not prioritize pure inference speed.
>
> **Following your answer, we made some inference speed improvements** that were not currently implemented in the inference script. They mainly consist of slightly changing the attention computation so that a full mask is used—full attention between training embeddings and [training, testing] embeddings—instead of sparse attention between [training, testing] and [training, testing]. This allows us to use **flash-attention** (and we use **bf16**, which we were not using before for inference), resulting in a significant speed-up (about 2-3x). We also made use of multi-GPU on a single node (8 GPUs) instead of a single GPU.
>
> TabDPT (pure inference) would therefore be faster than a classical method if $N_{\text{test}} < N_{\text{train}} \times \text{factor}$, where
>
>
> $$\text{factor} = t_{train:classical} / (t_{test:TabDPT} - t_{test:classical})$$
>
> Using the Adult dataset as an example, and a single run of XGBoost: from the Tabzilla file, XGBoost trains in approximately 0.54s per 1000 samples, and inference is 0.023s per 1000 samples. TabDPT (context size 1024), with the new optimizations, has 0.37s per 1000 samples (previously 1.44s, **3.9x speedup**), and 0.79s for context size 2048 (previously 3.38s, **4.27x speedup**). We neglected index creation time for TabDPT which was 0.3s for 26k samples (0.011s/1000samples << 0.54), but we discuss the large dataset size with more complex indexes briefly below.
>
> Computing the factor for several algorithms:
>
> |                       | XGBoost | XGBoost (HPO) | CatBoost | CatBoost (HPO) | LightGBM | LightGBM (HPO) | FTTransformer | FTTransformer (HPO) |
> |-----------------------|---------|---------------|----------|----------------|----------|----------------|---------------|---------------------|
> | TabDPT (ctx=1024)     | 1.61    | 48.3          | 7.18     | 215.4          | 13.65    | 409.5          | 44.1          | 1323                |
> | TabDPT (ctx=2048)     | 0.71    | 21.3          | 3.01     | 90.3           | 5.66     | 169.8          | 12.92         | 387.6               |
>
> Where we used the training times in Section D.1 for FT-Transformer and Tabzilla values for training times and inference times for all the rest. For TabDPT, we averaged over 20 seeds.
>
> **We can see here that for TabDPT to be slower than a classical method, the test set needs to typically be larger than the training set, sometimes by orders of magnitude depending on the method and whether it is HPO tuned. Thus, in many real scenarios, TabDPT would be faster (once we have the base model).** Furthermore many scenarios only require a minimum speed which we think TabDPT could satisfy for various applications.
>
>
> Note that currently, the model is not even compiled because we are not using `nn.DistributedDataParallel`—which is more efficient but more complex to set up than nn.DataParallel—which would result in another speedup. We are not confident we can debug this before the end of the discussion period, but this will be done.
>
> Of course, for very large datasets, we would need to train an index in `faiss`, which would be akin to a training time for TabDPT; however, we think it is fair to think of TabDPT as a method similar in terms of tradeoffs to kNN, which is very widely used in industry.

---

> ### Author Response · Authors · 2024-12-03
> **Answer (2/2): Scaling laws, contribution and score**
>
> ### Contribution and Scaling Laws
>
> We are pleased to see you mention the scaling laws as a contribution.
>
> While we agree that there has been work on **LLM-based models trained on real data (such as Tabula8B, NeurIPS 2024)** and tabular foundation models (TFM) trained on synthetic data, **the former have low performance (see, for instance, our 100% win rate ratio against Tabula8B). Our work actually goes counter to the latter trend in tabular foundation models, which use synthetic data ([TabPFN](https://arxiv.org/abs/2207.01848), [ForestPFN](https://arxiv.org/abs/2405.13396), [Attic](https://openreview.net/forum?id=DSl9sSuUhp)); we show improved performance when properly leveraging real data.**
>
> We think our work quantitatively answers the following questions: 1) How much real data is needed to train a strong model? 2) How can we use it efficiently? 3) Will tabular data unlock scaling like text data did?
>
> We believe our quantitative analysis addresses these open questions. We show that it is indeed possible to train a TFM with only 123 datasets. When using only 62 of these datasets (the 598M parameter line in Figure 1), we can outperform our improved version of TabPFN trained on synthetic data. However, to do this, it is important to make the most of the training data using the methods we described (column shuffling, masking, retrieval).
>
> Finally, we showed that this procedure allows us to scale model performance with the amount of data, similar to text or image data.  **Note that the paper "Scaling Laws for Time Series" recently provided scaling law analysis for time series and is currently very well received at ICLR 2025, showing that there is a strong interest in scaling laws for other data modalities** ([link](https://openreview.net/forum?id=uCqxDfLYrB)).
>
> Concerning the comment on ChatGPT, we are at the beginning of the large TFM era, and we are among the first to quantify scaling laws for tabular data, which are extremely important when developing such models. While we are not releasing the GPT-4 of tabular data—to keep your comparison—our work could be more on the timeline between GPT-1 (needed to be fine-tuned to be competitive) and GPT-2 (SOTA on many tasks without fine-tuning).
>
> ### Score
>
> Lastly, from our perspective, you pointed out some limitations of the current model with respect to the performance/inference time trade-off—which we acknowledged and addressed in this message. You agree that the evaluation is solid, the model performs strongly, and that the research direction is great.  **However, you gave us a score of 3, which, if all reviewers were to agree with you, would place this paper in the bottom 5% of all active submissions.**
>
> Recommending acceptance of this paper would not mean you agree with all choices made in our paper, but merely that you think this work is worth being presented at a conference to the community.  **We think this work deserves to be presented at ICLR 2025 and shown to the community. Given the current scores, the paper may not even be considered borderline and may not be discussed among reviewers**, which we think is a disservice to not only ourselves but the community at large as well.

---

### Author Response · Authors · 2024-11-22
**Review Response Summary**

Thank you to all the reviewers for taking the time to give feedback on our paper. We will update the paper based on your suggestions, and we invite you all to please discuss our rebuttals to your thoughtful reviews.

We would like to also briefly discuss **novelty** with the reviewers. TabDPT is the **first tabular model trained on real data providing quality predictions on completely unseen tasks with no further training**, representing strong novelty in our opinion. To do this we used techniques such as self-supervised learning, retrieval-based training, and a well-known transformer architecture; we will add extra citations on these points as noted by the reviewers, although our intention was never to claim novelty of these techniques.
**Our scaling analysis is also novel for tabular data**: scaling laws have been hugely important for the advancement of LLMs, and we are very eager to have discussions regarding this contribution.

Furthermore, following discussion with **Reviewer `h25x`**, we have discovered that TabDPT -- again with no further downstream training -- is **competitive with SOTA few-shot tabular techniques** like STUNT. This demonstrates the **foundational capability of TabDPT**, quickly delivering strong results in a novel setting.

---

> ### Author Response · Authors · 2024-12-04
> **Additional inference speed improvements**
>
> Following our conversation with **Reviewer** `HS6o`, we have optimized the attention mechanism, allowing us to use flash attention, and also added basic multi-GPU support. This resulted in a **speed-up of our inference by about 4x**. Along with our previous few-shot learning results, this further broadens the scope of application of our method, TabDPT.

---

### Meta-Review · Area_Chair_nhRt · 2024-12-19

**Metareview:**

This paper introduces TabDPT, which builds on TabPFN framework by training on real data, scaling model and dataset size, and introducing retrieval-based self-supervised learning techniques. The authors argue that their main contributions lie in the application of scaling laws for tabular data and the lessons learned from training large-scale tabular models. They also emphasize the novelty of transferring these models to real data, which contrasts with previous work that relies on synthetic datasets.

During the rebuttal period, the authors actively engaged in discussions with the reviewers, and the final scores were (8, 5, 5, 3). While the paper is clearly written and well-structured, with valuable contributions such as the scaling laws and practical insights from training on real tabular data, these contributions do not provide sufficient novelty or performance improvements over existing methods like TabPFN.

The final decision is reject for several reasons:
1. The core methodology, including retrieval and self-supervised learning, is not new and is already well-explored in tabular data. Furthermore, the performance improvements over TabPFN are incremental and do not justify the substantial increase in computational cost.
2. The claim of 'out-of-the-box' performance without fine-tuning is undermined by the model's reliance on fixed features and the limitations of inference time efficiency.
3. The model still has limitations on learning with larger datasets, i.e., those datasets with high-dimensional features and large class numbers.

**Additional Comments On Reviewer Discussion:**

The paper receives mixed scores, with one strong positive (8) and three negatives (5, 5, 3).

Reviewer HS6o mentions the limitations of the technical novelty and insignificant performance.

Reviewer 9jvJ has concerns about the scalability of TabDPT on large datasets, for example, with many classes and high dimensional features. The performance of vanilla TabPFN may vary when the number of ensembles is increased. Therefore, a comprehensive comparison between the proposed TabDPT and the ensemble version of TabPFN is necessary.

Reviewer jFpp also pointed out the limitation to deal with large classes and high-dimensional features. Applying PCA to high-dimensional features may influence the efficiency of the model, and the features extracted by PCA may not fit the pre-trained TabDPT.

Reviewer h25x gives positive scores, and the authors have addressed the concerns during the rebuttal period.

AC agrees with Reviewer 9jvJ and Reviewer jFpp on the potential issues of applying TabDPT on larger datasets, which is important for a general tabular model. The paper may need additional strategies to deal with the case. So the final decision is reject.

---

### Decision · Program_Chairs · 2025-01-22

Reject